# ERL-Re²: Efficient Evolutionary Reinforcement Learning with Shared State Representation and Individual Policy Representation

**Jianye Hao, Pengyi Li, Hongyao Tang, Yan Zheng, Xian Fu, Zhaopeng Meng**
College of Intelligence and Computing, Tianjin University, China

## Abstract

Deep Reinforcement Learning (Deep RL) and Evolutionary Algorithms (EA) are two major paradigms of policy optimization with distinct learning principles, i.e., gradient-based v.s. gradient-free. An appealing research direction is integrating Deep RL and EA to devise new methods by fusing their complementary advantages. However, existing works on combining Deep RL and EA have two common drawbacks: 1) the RL agent and EA agents learn their policies individually, neglecting efficient sharing of useful common knowledge; 2) parameter-level policy optimization guarantees no semantic level of behavior evolution for the EA side. In this paper, we propose **E**volutionary **R**einforcement **L**earning with Two-scale State **Re**presentation and Policy **Re**presentation (ERL-Re²), a novel solution to the aforementioned two drawbacks. The key idea of ERL-Re² is *two-scale* representation: all EA and RL policies *share* the same nonlinear state representation while maintaining *individual* linear policy representations. The state representation conveys expressive common features of the environment learned by all the agents collectively; the linear policy representation provides a favorable space for efficient policy optimization, where novel *behavior-level* crossover and mutation operations can be performed. Moreover, the linear policy representation allows convenient generalization of policy fitness with the help of the Policy-extended Value Function Approximator (PeVFA), further improving the sample efficiency of fitness estimation. The experiments on a range of continuous control tasks show that ERL-Re² consistently outperforms advanced baselines and achieves the State Of The Art (SOTA). Our code is available on https://github.com/yeshenpy/ERL-Re2.

## 1 Introduction

Reinforcement learning (RL) has achieved many successes in robot control (Yuan et al., 2022), game AI (Hao et al., 2022; 2019), supply chain (Ni et al., 2021) and etc (Hao et al., 2020). With function approximation like deep neural networks, the policy can be learned efficiently by trial-and-error with reliable gradient updates. However, RL is widely known to be unstable, poor in exploration, and struggling when the gradient signals are noisy and less informative. By contrast, Evolutionary Algorithms (EA) (Bäck & Schwefel, 1993) are a class of black-box optimization methods, which is demonstrated to be competitive with RL (Such et al., 2017). EA model natural evolution processes by maintaining a population of individuals and searching for favorable solutions by iteration. In each iteration, individuals with high fitness are selected to produce offspring by inheritance and variation, while those with low fitness are eliminated. Different from RL, EA are gradient-free and offers several strengths: strong exploration ability, robustness, and stable convergence (Sigaud, 2022). Despite the advantages, one major bottleneck of EA is the low sample efficiency due to the iterative evaluation of the population. This issue becomes more stringent when the policy space is large (Sigaud, 2022).

Since EA and RL have distinct and complementary advantages, a natural idea is to combine these two heterogeneous policy optimization approaches and devise better policy optimization algorithms. Many efforts in recent years have been made in this direction (Khadka & Tumer, 2018; Khadka et al., 2019; Bodnar et al., 2020; Wang et al., 2022; Shen et al., 2020). One representative work is ERL (Khadka & Tumer, 2018) which combines Genetic Algorithm (GA) (Mitchell, 1998) and DDPG (Lillicrap et al., 2016). ERL maintains an evolution population and a RL agent meanwhile. The population and the RL agent interact with each other in a coherent framework: the RL agent

learns by DDPG with diverse off-policy experiences collected by the population; while the population includes a copy of the RL agent periodically among which genetic evolution operates. In this way, EA and RL cooperate during policy optimization. Subsequently, many variants and improvements of ERL are proposed, e.g., to incorporate Cross-Entropy Method (CEM) (Pourchot & Sigaud, 2019) rather than GA (Pourchot & Sigaud, 2019), to devise gradient-based genetic operators (Gangwani & Peng, 2018), to use multiple parallel RL agents (Khadka et al., 2019) and etc. However, we observe that most existing methods seldom break the performance ceiling of either their EA or RL components (e.g., *Swimmer* and *Humanoid* on MuJoCo are dominated by EA and RL respectively). This indicates that the strengths of EA and RL are not sufficiently blended. We attribute this to two major drawbacks. First, each agent of EA and RL learns its policy individually. The state representation learned by individuals can inevitably be redundant yet specialized (Dabney et al., 2021), thus slowing down the learning and limiting the convergence performance. Second, typical evolutionary variation occurs at the level of the parameter (e.g., network weights). It guarantees no semantic level of evolution and may induce policy crash (Bodnar et al., 2020).

In the literature of linear approximation RL (Sutton & Barto, 1998) and state representation learning (Chung et al., 2019; Dabney et al., 2021; Kumar et al., 2021), a policy is usually understood as the composition of nonlinear state features and linear policy weights. Taking this inspiration, we propose a new approach named **E**volutionary **R**einforcement **L**earning with Two-scale State **Re**presentation and Policy **Re**presentation (ERL-Re$^2$) to address the aforementioned two drawbacks. ERL-Re$^2$ is devised based on a novel concept, i.e., *two-scale* representation: all EA and RL agents maintained in ERL-Re$^2$ are composed of a *shared* nonlinear state representation and an *individual* linear policy representation. The shared state representation takes the responsibility of learning general and expressive features of the environment, which is not specific to any single policy, e.g., the common decision-related knowledge. In particular, it is learned by following a unifying update direction derived from value function maximization regarding all EA and RL agents collectively. Thanks to the expressivity of the shared state representation, the individual policy representation can have a simple linear form. It leads to a fundamental distinction of ERL-Re$^2$: evolution and reinforcement occur in the linear policy representation space rather than in a nonlinear parameter (e.g., policy network) space as the convention. Thus, policy optimization can be more efficient with ERL-Re$^2$. In addition, we propose novel *behavior-level* crossover and mutation that allow to imposing variations on designated dimensions of action while incurring no interference on the others. Compared to parameter-level operators, our behavior-level operators have clear genetic semantics of behavior, thus are more effective and stable. Moreover, we further reduce the sample cost of EA by introducing a new surrogate of fitness, based on the convenient incorporation of Policy-extended Value Function Approximator (PeVFA) favored by the linear policy representations. Without loss of generality, we use GA and TD3 (and DDPG) for the concrete choices of EA and RL algorithms. Finally, we evaluate ERL-Re$^2$ on MuJoCo continuous control tasks with strong ERL baselines and typical RL algorithms, along with a comprehensive study on ablation, hyperparameter analysis, etc.

We summarize our major contributions below: 1) We propose a novel approach ERL-Re$^2$ to integrate EA and RL based on the concept of two-scale representation; 2) We devise behavior-level crossover and mutation which have clear genetic semantics; 3) We empirically show that ERL-Re$^2$ outperforms other related methods and achieves state-of-the-art performance.

## 2 BACKGROUND

**Reinforcement Learning** Consider a Markov decision process (MDP), defined by a tuple $\langle \mathcal{S}, \mathcal{A}, \mathcal{P}, \mathcal{R}, \gamma, T \rangle$. At each step $t$, the agent uses a policy $\pi$ to select an action $a_t \sim \pi(s_t) \in \mathcal{A}$ according to the state $s_t \in \mathcal{S}$ and the environment transits to the next state $s_{t+1}$ according to transition function $\mathcal{P}(s_t, a_t)$ and the agent receives a reward $r_t = \mathcal{R}(s_t, a_t)$. The return is defined as the discounted cumulative reward, denoted by $R_t = \sum_{i=t}^{T} \gamma^{i-t} r_i$ where $\gamma \in [0, 1)$ is the discount factor and $T$ is the maximum episode horizon. The goal of RL is to learn an optimal policy $\pi^*$ that maximizes the expected return. DDPG (Lillicrap et al., 2016) is a representative off-policy Actor-Critic algorithm, consisting of a deterministic policy $\pi_\omega$ (i.e., the actor) and a state-action value function approximation $Q_\psi$ (i.e., the critic), with the parameters $\omega$ and $\psi$ respectively. The critic is optimized with the Temporal Difference (TD) (Sutton & Barto, 1998) loss and the actor is updated by maximizing the estimated $Q$ value. The loss functions are defined as: $\mathcal{L}(\psi) = \mathbb{E}_\mathcal{D}[(r + \gamma Q_{\psi'}(s', \pi_{\omega'}(s')) - Q_\psi(s, a))^2]$ and $\mathcal{L}(\omega) = -\mathbb{E}_\mathcal{D}[Q_\psi(s, \pi_\omega(s))]$, where the experiences $(s, a, r, s')$ are sampled from the replay buffer $\mathcal{D}$, $\psi'$ and $\omega'$ are the parameters of the target networks. TD3 (Fujimoto et al., 2018) improves DDPG by addressing overestimation issue mainly by clipped double-$Q$ learning.

Conventional value functions are defined on a specific policy. Recently, a new extension called Policy-extended Value Function Approximator (PeVFA) (Tang et al., 2022) is proposed to preserve the values of multiple policies. Concretely, given some representation $\chi_\pi$ of policy $\pi$, a PeVFA parameterized by $\theta$ takes as input $\chi_\pi$ additionally, i.e., $\mathbb{Q}_\theta(s, a, \chi_\pi)$. Through the explicit policy representation $\chi_\pi$, one appealing characteristic of PeVFA is the value generalization among policies (or policy space). PeVFA is originally proposed to leverage the local value generalization along the policy improvement path to improve online RL (we refer the reader to their original paper). In our work, we adopt PeVFA for the value estimation of the EA population, which naturally fits the ability of PeVFA well. Additionally, different from the on-policy learning of PeVFA adopted in (Tang et al., 2022), we propose a new off-policy learning algorithm of PeVFA which is described later.

**Evolutionary Algorithm** Evolutionary Algorithms (EA) (Bäck & Schwefel, 1993) are a class of black-box optimization methods. EA maintains a population of policies $\mathbb{P} = \{\pi_1, \pi_2, ..., \pi_n\}$ in which policy evolution is iteratively performed. In each iteration, all agents interact with the environment for $e$ episodes to obtain Monte Carlo (MC) estimates of policy fitness $\{f(\pi_1), f(\pi_2), ..., f(\pi_n)\}$ where $f(\pi) = \frac{1}{e} \sum_{i=1}^{e} [\sum_{t=0}^{T} r_t \mid \pi]$. The policy with higher fitness is more likely to be selected as parents to produce the next generation in many ways such as Genetic Algorithm (GA) (Mitchell, 1998) and Cross-Entropy Method (CEM) (Pourchot & Sigaud, 2019). With GA, offspring are generated by applying genetic operators: the parents $\pi_i$ and $\pi_j$ are selected randomly to produce offspring $\pi_i'$ and $\pi_j'$ by performing the crossover operator, i.e., $\pi_i', \pi_j' = \texttt{Crossover}(\pi_i, \pi_j)$ or the mutation operator $\pi_i' = \texttt{Mutation}(\pi_i)$. In most prior methods, the crossover and mutation operate at the *parameter level*. Typically, $k$-point crossover randomly exchange segment-wise (network) parameters of parents while Gaussian mutation adds Gaussian noises to the parameters. Thanks to the diversity brought by abundant candidates and consistent variation, EA has strong exploration ability and convergence.

## 3 RELATED WORK

Recently, an emergent research field is integrating the advantages of EA and RL from different angles to devise new methods (Sigaud, 2022), for example, combining EA and RL for efficient policy optimization (Khadka & Tumer, 2018; Bodnar et al., 2020), using EA to approximate the greedy action selection in continuous action space (Kalashnikov et al., 2018; Simmons-Edler et al., 2019; Shi & Singh, 2021; Shao et al., 2021; Ma et al., 2022), population-based hyperparameter tuning of RL (Jaderberg et al., 2017; Pretorius & Pillay, 2021), and genetic programming for interpretable RL policies (Hein et al., 2018; Hein, 2019). In this work, we focus on combining EA and RL for efficient policy optimization. ERL (Khadka & Tumer, 2018) first proposes a hybrid framework where a DDPG agent is trained alongside a genetic population. The RL agent benefits from the diverse experiences collected by the EA population, while the population periodically includes a copy of the RL agent. In parallel, CEM-RL (Pourchot & Sigaud, 2019) integrates CEM and TD3. In particular, the critic function of TD3 is used to provide update gradients for half of the individuals in the CEM population. Later, ERL serves as a popular framework upon which many improvements are made. CERL (Khadka et al., 2019) extends the single RL agent to multiple ones with different hyperparameter settings to make better use of the RL side. GPO (Gangwani & Peng, 2018) devises gradient-based crossover and mutation by policy distillation and policy gradient algorithms, respectively. Further, PDERL (Bodnar et al., 2020) devises the $Q$-filtered distillation crossover and Proximal mutation to alleviate the policy crash caused by conventional genetic operators at the parameter level. All these works adopt no sharing among agents and each agent of EA and RL learns its own state representation, which is inefficient and specialized. By contrast, our approach ERL-Re$^2$ makes use of a expressive state representation function which is shared and learned by all agents. Another common point of these works is, evolution variations are imposed at the parameter level (i.e., policy network). Despite the existence of GPO and PDERL, the semantics of genetic operators on policy behavior cannot be guaranteed due to the nonlinearity nature of policy parameters. In ERL-Re$^2$, we propose behavior-level crossover and mutation operators that have clear semantics with linear policy representations.

## 4 REPRESENTATION-BASED EVOLUTIONARY REINFORCEMENT LEARNING

In this section, we introduce the overview of ERL-Re$^2$ to gain the holistic understanding of the key concept. In addition, we introduce how ERL-Re$^2$ can be realized by a general form of algorithm framework. We defer the concrete implementation details in Sec. 5.

## 4.1 THE CONCEPT OF TWO-SCALE STATE REPRESENTATION AND POLICY REPRESENTATION

The previous algorithms that integrate EA and RL for policy optimization primarily follow the ERL interaction architecture shown on the left of Fig. 1, where the population's policies offer a variety of experiences for RL training and the RL side injects its policy into the population to participate in the iterative evolution. However, two significant issues exist: 1) each agent maintains an independent nonlinear policy and searches in parameter space, which is inefficient because each agent has to independently and repeatedly learn common and useful knowledge. 2) Parameter-level perturbations might result in catastrophic failures, making parameter-level evolution exceedingly unstable.

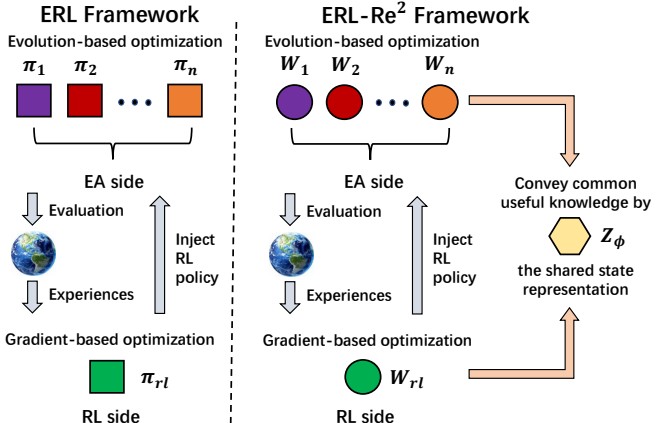

Figure 1: The left represents ERL framework and the right represents ERL-Re$^2$ framework. In ERL-Re$^2$, all the policies are composed of the nonlinear shared state representation $Z_\phi$ and an individual linear policy representation $W$.

To address the aforementioned problems, we propose Two-scale Representation-based policy construction, based on which we maintain and optimize the EA population and RL agent. The policy construction is illustrated on the right of Fig. 1. Specifically, the policies for EA and RL agents are all composed of a *shared* nonlinear state representation $z_t = Z_\phi(s_t) \in \mathbb{R}^d$ (given a state $s_t$) and an *individual* linear policy representation $W \in \mathbb{R}^{(d+1) \times |\mathcal{A}|}$. We refer to the different representation scopes (shared/individual + state/policy representation) of the policy construction as the two scales. The agent $i$ makes decisions by combining the shared state representation and the policy representation:

$$\pi_i(s_t) = \text{act}(Z_\phi(s_t)^\mathsf{T} W_{i,[1:d]} + W_{i,[d+1]}) \in \mathbb{R}^{|\mathcal{A}|},$$

where $W_{[m(:n)]}$ denotes the slice of matrix $W$ that consists of row $m$ (to $n$) and $\text{act}(\cdot)$ denotes some activation function (e.g., $\text{tanh}$)[1]. In turn, we also denote the EA population by $\mathbb{P} = \{W_1, W_2, ..., W_n\}$ and the RL agent by $W_{rl}$. Intuitively, we expect the shared state representation $Z_\phi$ to be useful to all possible policies encountered during the learning process. It ought to contain the general decision-related features of the environment, e.g., common knowledge, while not specific to any single policy. By sharing the state representation $Z_\phi$, it does not require each agent to learn how to represent the state independently. Thus, higher efficiency and more expressive state representation can be fulfilled through learning in a collective manner with the EA population and RL agent. Since $Z_\phi$ is responsible for expressivity, each individual policy representation can have a straightforward linear form that is easy to optimize and evaluate (with PeVFA).

## 4.2 THE ALGORITHM FRAMEWORK OF ERL-RE$^2$

Due to the representation-based policy construction, the state representation function $Z_\phi$ determines a policy space denoted by $\Pi(\phi)$, where we optimize the individual representations of the EA and RL agents. The optimization flow of ERL-Re$^2$ is shown in Fig. 2. The top and bottom panels depict the learning dynamics of $Z_\phi$ and the agents, respectively. In each iteration $t$, the agents in the EA population $\mathbb{P}$ and the RL agent $W_{rl}$ evolve or reinforce their representations in the policy space $\Pi(\phi_t)$ provided by $Z_{\phi_t}$ (Sec. 5.2). After the optimization at the scale of individual representation, the shared state representation is optimized (i.e., $\phi_t \rightarrow \phi_{t+1}$) towards a unifying direction, derived from value function maximization regarding all the EA and RL agents (Sec. 5.1). By this means, the shared state representation is optimized in the direction of a superior policy space for successive policy optimization. In an iterative manner, the shared state representation and the individual policy representations play distinct roles and complement each other in optimizing the EA and RL agents.

In principle, ERL-Re$^2$ is a general framework that can be implemented with different EA and RL algorithms. For the side of EA, we mainly consider Genetic Algorithm (GA) (Mitchell, 1998)

---

[1]We use deterministic policy for demonstration while the construction is compatible with stochastic policy.

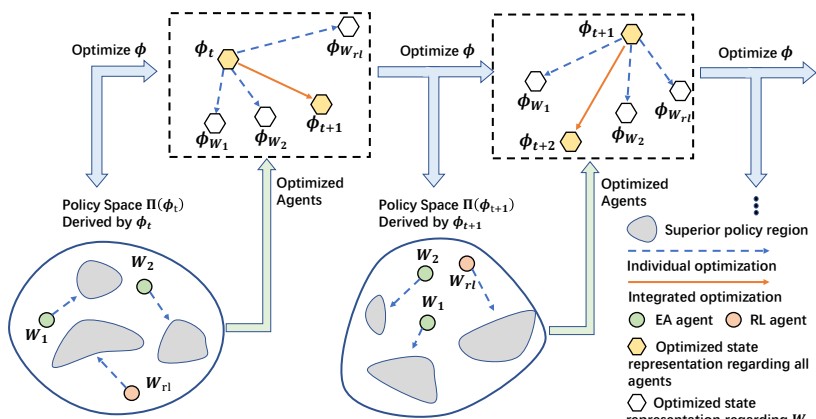

Figure 2: The optimization flow of ERL-Re$^2$. In an iterative fashion, the shared state representation and individual linear policy representations are optimized at two scales.

as a representative choice while another popular choice CEM (Pourchot & Sigaud, 2019) is also discussed in Appendix F. Our linear policy representation can realized the genetic operations with clear semantics. For the side of RL, we use DDPG (Lillicrap et al., 2016) and TD3 (Fujimoto et al., 2018). For efficient knowledge sharing and policy optimization, we learn a shared state representation by all the agents, then we evolve and reinforce the policies in the policy representation space rather than in the original policy parameter space.

A general pseudo-code of ERL-Re$^2$ is shown in Algorithm 1. In each iteration, the algorithm proceeds across three phases (denoted by blue). First, each agent of EA and RL interacts with the environment and collects the experiences. Specifically, the agents in the EA population $\mathbb{P}$ have the probability $1 - p$ to rollout partially and estimate the surrogate fitness with higher sample efficiency (Line 5-6), in contrast to the conventional way (Line 7-8). Next, evolution and reinforcement update /optimize their policy representations in the linear policy space (Line 13-14). The agents are optimized by EA and RL, where RL agent learns with additional off-policy experiences collected by the agents in $\mathbb{P}$ (Line 14) and periodically injects its policy to $\mathbb{P}$ (Line 15). Finally, the shared state representation is updated to provide superior policy space for the following iteration (Line 17).

## 5 EVOLUTION AND REINFORCEMENT IN REPRESENTATION SPACE

In this section, we present the algorithm details of how to optimize the two-scale representation. In addition, we introduce a new surrogate of policy fitness to further improve the sample efficiency.

### 5.1 OPTIMIZING THE SHARED STATE REPRESENTATION FOR A SUPERIOR POLICY SPACE

The shared state representation $Z_\phi$ designates the policy space $\Pi_\phi$, in which EA and RL are conducted. As discussed in the previous section, the shared state representation takes the responsibility of learning useful features of the environment from the overall policy learning. We argue that this is superior to the prior way, i.e., each agent learns its individual state representation which can be less efficient and limited in expressivity. In this paper, we propose to learn the shared state representation by the principle of *value function maximization regarding all EA and RL agents*. To be specific, we first detail the value function approximation for both EA and RL agents. For the EA agents, we learn a PeVFA $\mathbb{Q}_\theta(s, a, W_i)$ based on the linear policy representation $W_i$ in EA population $\mathbb{P}$; for the RL agent, we learn a critic $Q_\psi(s, a)$. In principle, RL can use PeVFA as its critic. We experimentally show that both approaches can achieve similar performance in Appendix.D.4. The loss functions of $\mathbb{Q}_\theta$ and $Q_\psi$ are formulated below:

$$\mathcal{L}_\mathbb{Q}(\theta) = \mathbb{E}_{(s,a,r,s')\sim\mathcal{D}, W_i \sim \mathbb{P}} \left[ \left( r + \gamma \mathbb{Q}_{\theta'} \left( s', \pi_i(s'), W_i \right) - \mathbb{Q}_\theta \left( s, a, W_i \right) \right)^2 \right],$$

$$\mathcal{L}_Q(\psi) = \mathbb{E}_{(s,a,r,s')\sim\mathcal{D}} \left[ \left( r + \gamma Q_{\psi'} \left( s', \pi'_{rl}(s') \right) - Q_\psi \left( s, a \right) \right)^2 \right],$$

(1)

where $D$ is the experience buffer collected by both the EA and RL agents, $\theta', \psi'$ denote the target networks of the PeVFA and the RL critic, $\pi'_{rl}$ denote the target actor with policy representation $W'_{rl}$, and recall $\pi_i(s) = \mathrm{act}(Z_\phi(s)^\mathsf{T} W_{i,[1:d]} + W_{i,[d+1]}) \in \mathbb{R}^{|\mathcal{A}|}$. Note that we make $\mathbb{Q}_\theta$ and $Q_\psi$ take the raw state $s$ as input rather than $Z_\phi(s)$. Since sharing state representation between actor and critic

---

**Algorithm 1:** ERL with Two-scale State Representation and Policy Representation (ERL-Re$^2$)

---

1 **Input:** the EA population size $n$, the probability $p$ of using MC estimate, the partial rollout length $H$
2 **Initialize:** a replay buffer $\mathcal{D}$, the shared state representation function $Z_\phi$, the RL agent $W_{rl}$, the EA
    population $\mathbb{P} = \{W_1, \cdots, W_n\}$, the RL critic $Q_\psi$ and the PeVFA $\mathbb{Q}_\theta$ (target networks are omitted here)
3 **repeat**
4     # Rollout the EA and RL agents with $Z_\phi$ and estimate the (surrogate) fitness
5     **if** *Random Number $> p$* **then**
6         Rollout each agent in $\mathbb{P}$ for $H$ steps and evaluate its fitness by the surrogate $\hat{f}(W)$     $\triangleright$ see Eq. 3
7     **else**
8         Rollout each agent in $\mathbb{P}$ for one episode and evaluate its fitness by MC estimate
9     Rollout the RL agent for one episode
10     Store the experiences generated by $\mathbb{P}$ and $W_{rl}$ to $\mathcal{D}$
11     # *Individual* scale: evolution and reinforcement in the policy space
12     Train PeVFA $\mathbb{Q}_\theta$ and RL critic $Q_\psi$ with $D$     $\triangleright$ see Eq. 1
13     **Optimize the EA population:** perform the genetic operators (i.e., selection, crossover and mutation) at
        the behavior level for $\mathbb{P} = \{W_1, \cdots, W_n\}$     $\triangleright$ see Eq. 4
14     **Optimize the RL agent:** update $W_{rl}$ (by e.g., DDPG and TD3) according to $Q_\psi$     $\triangleright$ see Eq. 5
15     Inject RL agent to the population $\mathbb{P}$ periodically
16     # *Common* scale: improving the policy space through optimizing $Z_\phi$
17     **Update the shared state representation:** optimize $Z_\phi$ with a unifying gradient direction derived from
        value function maximization regarding $\mathbb{Q}_\theta$ and $Q_\psi$     $\triangleright$ see Eq. 2
18 **until** *reaching maximum steps*;

---

may induce interference and degenerated performance as designated by recent studies (Cobbe et al., 2021; Raileanu & Fergus, 2021). Another thing to notice is, to our knowledge, we are the first to train PeVFA in an off-policy fashion (Eq. 1). We discuss more on this in Appendix F.

For each agent of EA and RL, an individual update direction of the shared state representation $Z_\phi$ is now ready to obtain by $\nabla_\phi \mathbb{Q}_\theta(s, \pi_i(s), W_i)$ for any $W_i \in \mathbb{P}$ or $\nabla_\phi Q_\psi(s, \pi_{rl}(s))$ through $\pi_i$ and $\pi_{rl}$ respectively. This is the value function maximization principle where we adjust $Z_\phi$ to induce superior policy (space) for the corresponding agent. To be *expressive*, $Z_\phi$ should not take either individual update direction solely; instead, the natural way is to take a unifying update direction regarding all the agents (i.e., the currently *targeted* policies). Finally, the loss function of $Z_\phi$ is defined:

$$\mathcal{L}_Z(\phi) = -\mathbb{E}_{s \sim \mathcal{D}, \{W_j\}_{j=1}^K \sim \mathbb{P}} \Big[ Q_\psi\left(s, \pi_{rl}\left(s\right)\right) + \sum_{j=1}^K \mathbb{Q}_\theta\left(s, \pi_j\left(s\right), W_j\right) \Big], \tag{2}$$

where $K$ is the size of the sampled subset of EA agents that engage in updating $Z_\phi$. By minimizing Eq. 2, the shared state representation $Z_\phi$ is optimized towards a superior policy space $\Pi_\phi$ pertaining to all the EA and RL agents iteratively, as the learning dynamics depicted in Fig. 2. Note that the value maximization is not the only possible choice. For a step further, we also investigate on the incorporation of self-supervised state representation learning in Appendix D.

## 5.2 Optimizing the Policy Representation by Evolution and Reinforcement

Given the shared state representation $Z_\phi$, all the agents of EA and RL optimize their individual policy representation in the policy space $\Pi(\phi)$. The fundamental distinction here is, that the evolution and reinforcement occur in the linear policy representation space rather than in a nonlinear policy network parameter space as the convention. Thus, policy optimization can be efficiently conducted. Moreover, the linear policy representation has a special characteristic: it allows performing genetic operations at the behavior level. We detail the policy representation optimization of EA and RL below.

The evolution of the EA population $\mathbb{P}$ mainly consists of: 1) interaction and selection, 2) genetic evolution. For the process of interaction and selection, most prior methods rollout each agent in $\mathbb{P}$ for one or several episodes and calculate the MC fitness. The incurred sample cost is regarded as one major bottleneck of EA especially when the population is large. To this end, we propose a new surrogate fitness based on the PeVFA $\mathbb{Q}_\theta$ (Eq. 1) and partial rollout. At the beginning of each iteration, we have a probability $p$ to rollout the EA population for $H$ steps. For each agent $W_i$, the surrogate fitness is estimated by $H$-*step bootstrapping (Sutton & Barto, 1998)*:

$$\hat{f}(W_i) = \sum_{t=0}^{H-1} \gamma^t r_t + \gamma^H \mathbb{Q}_\theta(s_H, \pi_i(s_H), W_i). \tag{3}$$

Thanks to the PeVFA $\mathbb{Q}_\theta$, the surrogate fitness can be conveniently estimated and reduce the sample cost effectively. Moreover, one key point is that $\hat{f}(W_i)$ is free of the off-policy bias in the surrogate proposed in the concurrent work (Wang et al., 2022). To be concrete, the proposed surrogate fitness for EA agents in (Wang et al., 2022) is estimated by bootstrapping based on the RL critic and uniformly samples from the replay buffer. Apparently, the off-policy bias in their surrogate comes from the mismatch of value functions between specific EA agents and the RL agent, and the mismatch between initial state distribution and the replay buffer. By contrast, our surrogate fitness obviates such mismatches by bootstrapping from PeVFA and using partial rollout. We experimentally prove that $\hat{f}(W_i)$ is more efficient. Therefore, according to the fitness estimated by MC or our surrogate, parents with high fitness are more likely to be selected as parents. The details of the selection process follow PDERL (Bodnar et al., 2020).

For the process of genetic evolution, two genetic operators are involved: crossover and mutation. Typical $k$-point crossover and Gaussian mutation that directly operate at the parameter level can easily lead to large fluctuations in the decision and even cause a crash to policy (Gangwani & Peng, 2018; Bodnar et al., 2020). The reason behind this is that the policy parameter is highly nonlinear to behavior, thus the natural semantics could be hardly guaranteed by such parameter-level operators. In contrast, recall the linear policy representation $W$ in a matrix form. Each row $W_{[i]}$ determines the $i$-th dimension of action and thus the change to $W_{[i]}$ does not affect the behavior of other dimensions. Based on the granularity of clearer semantics, we propose novel *behavior-level* crossover and mutation, as the illustration in Fig.3. For the behavior-level crossover ($b$-Crossover), the offspring are produced by inheriting the behaviors of the parents at the corresponding randomly selected dimensions. For the behavior-level mutation ($b$-Mutation), each dimension of behavior has the probability $\alpha$ to be perturbed. Formally, we formulate the two operations below:

$$(W_{c_1}, W_{c_2}) = (W_{p_1} \otimes_{d_1} W_{p_2}, W_{p_2} \otimes_{d_2} W_{p_1}) = b\text{-Crossover}(W_{p_1}, W_{p_2}),$$
$$W_{m_1} = W_{p_1} \otimes_{\hat{d}_1} P_1 = b\text{-Mutation}(W_{p_1}),$$
(4)

where $d_1, d_2, \hat{d}_1$ are the randomly sampled subsets of all dimension indices, $P_1$ is the randomly generated perturbation matrix, and we use $A \otimes_d B$ to denote $A$ being replaced by $B$ at the corresponding rows in $d$. For the dimension selection and perturbation generation, we follow the same approach used in (Bodnar et al., 2020) (detailed in Appendix C) and we analyze the hyperparameter choices specific to our approach in Sec. 6.3. By imposing variations on specific behavior dimensions while incurring no interference on the others, our proposed operators (i.e., $b$-Crossover and $b$-Mutation) can be more effective and stable, as well as more informative in the sense of genetic semantics.

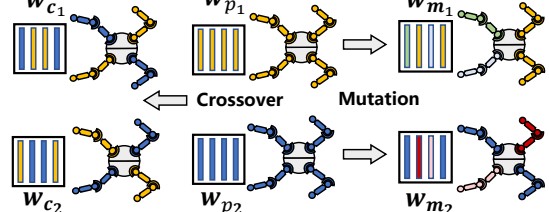

Figure 3: Behavior-level crossover and mutation operators in ERL-Re$^2$. The offspring and parents are indexed by $c_1, c_2, m_1, m_2$ and $p_1, p_2$ respectively.

As to the learning of the RL agent $W_{rl}$, it resembles the conventional circumstance except done with respect to the linear policy representation. Taking DDPG (Lillicrap et al., 2016) for a typical example, the loss function of $W_{rl}$ is defined below, based on the RL critic $Q_\psi$ (learned by Eq. 1):

$$\mathcal{L}_{\text{RL}}(W_{rl}) = -\mathbb{E}_{s \sim \mathcal{D}}\Big[Q_\psi\left(s, \pi_{rl}(s)\right)\Big].$$
(5)

The RL agent learns from the off-policy experience in the buffer $D$ collected also by the EA population. Meanwhile, the EA population incorporates the RL policy representation $W_{rl}$ at the end of each iteration. By such an interaction, EA and RL complement each other consistently and the respective advantages of gradient-based and gradient-free policy optimization are merged effectively.

# 6 EXPERIMENTS

## 6.1 EXPERIMENTAL SETUPS

We evaluate ERL-Re$^2$ on six MuJoCo (Todorov et al., 2012) continuous control tasks as commonly used in the literature: *HalfCheetach*, *Swimmer*, *Hopper*, *Ant*, *Walker*, *Humanoid* (all in version 2). For a comprehensive evaluation, we implement two instances of ERL-Re$^2$ based on DDPG (Lillicrap et al., 2016) and TD3 (Fujimoto et al., 2018), respectively. We compare ERL-Re$^2$ with the following

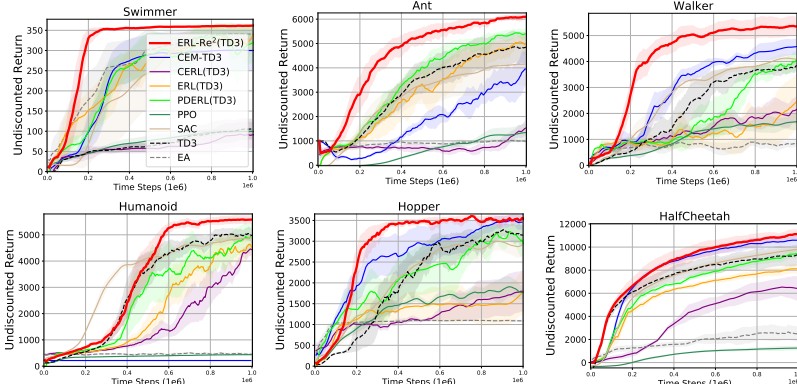

Figure 4: Performance comparison between ERL-Re$^2$ and baselines (all in the **TD3** version).

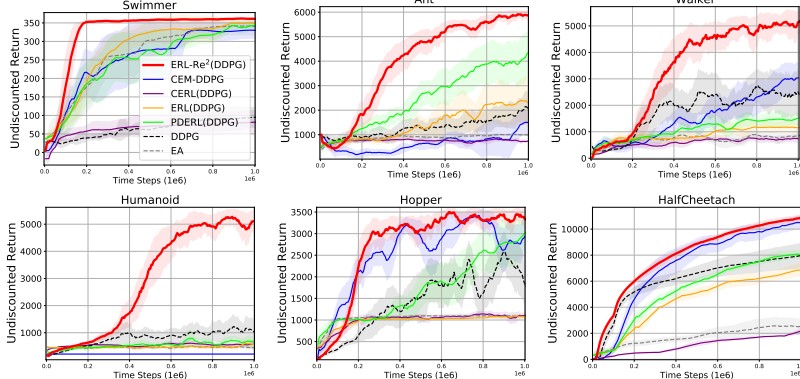

Figure 5: Performance comparison between ERL-Re$^2$ and baselines (all in the **DDPG** version).

baselines: 1) **basic baselines**, i.e., PPO (Schulman et al., 2017), SAC (Haarnoja et al., 2018), DDPG, TD3 and EA (Mitchell, 1998); 2) **ERL-related baselines**, including ERL (Khadka & Tumer, 2018), CERL (Khadka et al., 2019), PDERL (Bodnar et al., 2020), CEM-RL (Pourchot & Sigaud, 2019). The ERL-related baselines are originally built on different RL algorithms (either DDPG or TD3), thus we modify them to obtain both the two versions for each of them. We use the official implementation and *stable-baseline3* for the mentioned baselines in our experiments. For a fair comparison, we compare the methods built on the same RL algorithm (i.e., DDPG and TD3) and fine-tune them in each task to provide the best performance. All statistics are obtained from 5 independent runs. This is consistent with the setting in ERL and PDERL. We report the average with 95% confidence regions. For the population size $n$, we consider the common choices and use the best one in $\{5, 10\}$ for each concerned method. We implement our method ERL-Re$^2$ based on the codebase of PDERL and the common hyperparameters remain the same. For the hyperparameters specific to ERL-Re$^2$ (both DDPG and TD3 version), we set $\alpha$ to 1.0 and select $H$ from $\{50, 200\}$, $K$ from $\{1, 3\}$, $p$ from $\{0.3, 0.5, 0.7, 0.8\}$ for different tasks. All implementation details are provided in Appendix E.

## 6.2 PERFORMANCE EVALUATION

We evaluate ERL-Re$^2$ and baselines in TD3 and DDPG versions separately. The results in Fig.4 and Fig.5 show that both ERL-Re$^2$ (TD3) and ERL-Re$^2$ (DDPG) significantly outperform other methods in most tasks. It is worth noting that, to our knowledge, ERL-Re$^2$ is the first algorithm that outperforms EA in *Swimmer*. We can see that both ERL-Re$^2$ (TD3) and ERL-Re$^2$ (DDPG) achieve a 5x improvement in convergence rate compared to EA and obtain higher and more stable performance. Moreover, in some more difficult environments such *Humanoid* which is dominated by RL algorithms, ERL-Re$^2$ also achieves significant improvements. Overall, ERL-Re$^2$ is an effective and general framework that significantly improves both EA and RL in all the six tasks while other methods fails. Beyond MuJoCo, we also demonstrate the performance improvement achieved by ERL-Re$^2$ in several visual control tasks of DMC (Tassa et al., 2018) in Appendix D.5.

## 6.3 SUPERIORITY OF COMPONENTS & PARAMETER ANALYSIS

We conduct experiments to compare our proposed genetic operators and surrogate fitness with other related methods. In Fig.6a, we compare the behavior-level operators with other alternatives. The

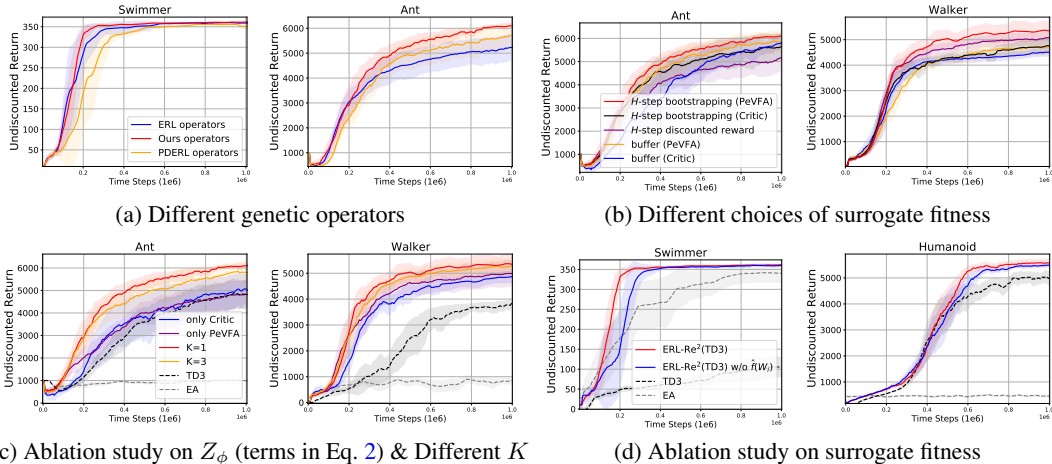

(a) Different genetic operators      (b) Different choices of surrogate fitness

(c) Ablation study on $Z_\phi$ (terms in Eq. 2) & Different $K$      (d) Ablation study on surrogate fitness

Figure 6: Comparative evaluation of the components in ERL-Re$^2$ and ablation study.

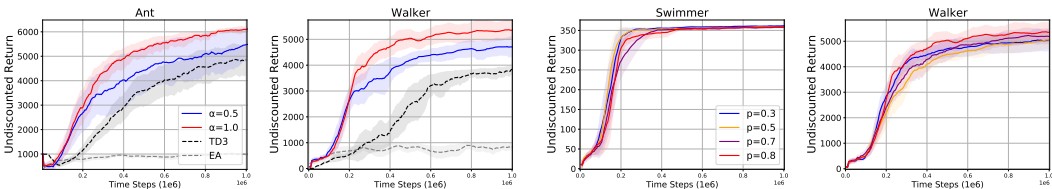

(a) Probability $\alpha$ of each action mutation      (b) Probability $p$ to use our surrogate fitness

Figure 7: Analysis of main hyperparameters.

results show that the behavior-level operators are more effective than other operators. Next, we examine whether the surrogate fitness estimate $\hat{f}(W_i)$ (i.e., $H$-step bootstrapping (PeVFA)) is better than using the RL critic to evaluate fitness based on the latest state from replay buffer (Wang et al., 2022) (i.e., buffer (Critic)). The results in Fig.6b show that using PeVFA is more effective than using the RL critic. This is because the RL critic cannot elude the off-policy bias while PeVFA is free of it thanks to the generalized approximation. Second, we provide ablation study on how to update $Z_\phi$ and analyze the effect of different $K$. We evaluate ERL-Re$^2$ (TD3) with different $K \in [1, 3]$ and ERL-Re$^2$ (TD3) with only PeVFA or critic to optimize the shared state representation. The results in Fig.6c demonstrate the conclusion of two aspects: 1) $K$ affects the results and appropriate tuning are beneficial; 2) both the RL critic and PeVFA play important roles. Only using the RL critic or PeVFA leads to inferior shared state representation, crippling the overall performance. Third, we perform ablation experiments on $\hat{f}(W_i)$. The results in Fig.6d show that using $\hat{f}(W_i)$ can deliver performance gains and achieve further improvement to ERL-Re$^2$ especially in *Swimmer*.

For hyperparameter analysis, the results in Fig.7a show that larger $\alpha$ is more effective and stable, which is mainly since that larger $\alpha$ can lead to stronger exploration. Thus we use $\alpha = 1.0$ on all tasks. Finally, the results in Fig. 7b demonstrate that $p$ has an impact on performance and appropriate tuning for different tasks is necessary. It is worth noting that setting $p$ to 1.0 also gives competitive performances, i.e., do not use the surrogate fitness $\hat{f}(W_i)$ (see blue in Fig. 6d).

**Others** Due to space limitations, more experiments on the hyperparameter analysis of $H$, $\beta$, different combinations of genetic operators, incorporating self-supervised state representation learning, the interplay between the EA population and the RL agent and etc. are placed in Appendix D.

## 7 CONCLUSION

We propose a novel approach called ERL-Re$^2$ to fuse the distinct advantages of EA and RL for efficient policy optimization. In ERL-Re$^2$, all agents are composed of the shared state representation and an individual policy representation. The shared state representation is endowed with expressive and useful features of the environment by value function maximization regarding all the agents. For the optimization of individual policy representation, we propose behavior-level genetic operators and a new surrogate of policy fitness to improve effectiveness and efficiency. In our experiments, we have demonstrated the significant superiority of ERL-Re$^2$ compared with various baselines.

ACKNOWLEDGMENTS

This work is supported by the National Natural Science Foundation of China (Grant No.62106172), the "New Generation of Artificial Intelligence" Major Project of Science & Technology 2030 (Grant No.2022ZD0116402), and the Science and Technology on Information Systems Engineering Laboratory (Grant No.WDZC20235250409, No.WDZC20205250407).

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

## A    LIMITATIONS & FUTURE WORK

For limitations and future work, firstly our work is empirical proof of the effectiveness of the ERL-Re$^2$ idea and we provide no theory on optimality, convergence, and complexity. Secondly, one critical point is the expressivity of the shared state representation, which is not optimally addressed by this work. Self-supervised state representation learning and nonlinear policy representation with behavior semantics can be potential directions. Thirdly, diversity is one key aspect of EA while in ERL-Re$^2$ we currently do not consider an explicit mechanism or objective to optimize the diversity of EA population (and also RL policy). One important future work is to integrate Quality-Diversity principle (Parker-Holder et al., 2020; Fontaine & Nikolaidis, 2021; Wang et al., 2021) for further development of ERL-Re$^2$. Fourthly, most current ERL methods including our ERL-Re$^2$ are model-free. Obviously, the model-free paradigm limits the advance in sample efficiency. One promising direction can be ERL with a world model, as a pioneer work (Ha & Schmidhuber, 2018) has demonstrated the feasibility with CMA-ES. For the final one, we follow the convention in the literature and evaluate ERL-Re$^2$ on MuJoCo in the main text and additional tasks DMC (Tassa et al., 2018) with image inputs in Appendix D.5; while performance in other environments, e.g., Atari (Mnih et al., 2013), is not thoroughly studied.

## B    COMPARISON BETWEEN ERL AND ERL-RE$^2$

In this section, we compare ERL (Khadka & Tumer, 2018) and ERL-Re$^2$ in detail. The ERL framework is currently one popular framework to integrate EA and RL for policy optimization. We illustrate the difference between the two frameworks intuitively in Fig.8. On the foundation of the interaction schema between EA and RL in ERL, ERL-Re$^2$ makes all policies composed of the shared state representation and individual policy representations. Based on the shared state representation, useful common knowledge can be delivered efficiently among agents. Meanwhile, the linear policy representations allow for imposing the novel behavior-level crossover and mutation operations, which are more stable and effective.

We detail how to update the shared state representation by the principle of *value function maximization*. Here we illustrate the process in Fig. 9 to help the readers understand it more intuitively. After interacting with the environment for $t$ steps, we optimize the shared state representation $t$ times. In each update, we maximize $Q(s, a)$ of the RL agent on the one hand. On the other hand, we sample $K$ policies $\{W_1, ...W_K\}$ from the population to maximize $\mathbb{Q}(s, a, W_i)$. Finally, the shared state representation is optimized toward a unifying direction according to Eq. 2.

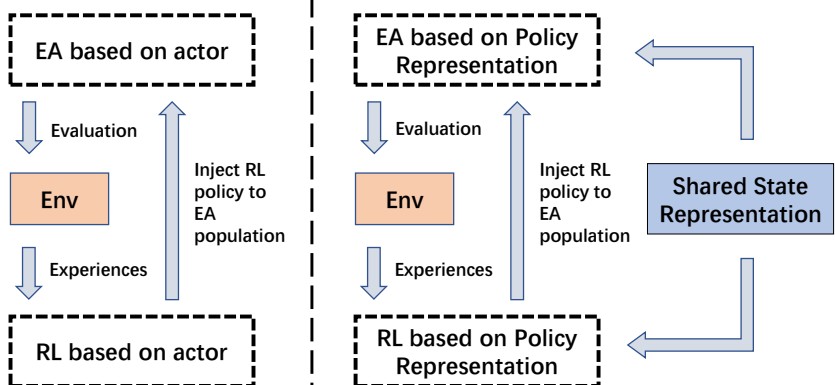

Figure 8: Comparison between ERL (Khadka & Tumer, 2018) and ERL-Re$^2$. The left shows the interaction between EA and RL in ERL, and the right shows the interaction between EA and RL in ERL-Re$^2$.

## C    DETAILS OF GENETIC ALGORITHM IN ERL-RE$^2$

In this section, we provide the genetic algorithm flow and the corresponding pseudo-code in Algorithm 2. Since ERL-Re$^2$ is built on the official code of PDERL (Bodnar et al., 2020), thus all hyperparameters and genetic algorithm flow remain the same.

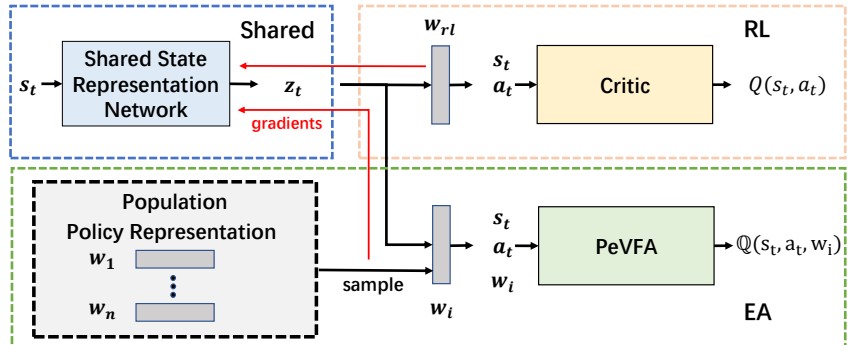

Figure 9: The illustration on how to update shared state representations. In each update, we sample $K$ policy representations from the population. Based on this, ERL-Re$^2$ updates the shared state representation network with PeVFA and critic according to Eq. 2.

Specifically, we first get the fitness of all policies in the population and divide all policies into three groups: elites, winners, and discarders. We choose the best $e$ policies as the elites `E` (In PDERL, $e$ is different across tasks. We choose 1 for all tasks in ERL-Re$^2$). Then we randomly select 3 policies from the non-elites policies and choose the best one as the winner. Then we repeat the process for $n - e$ times and get the set of these policies `Win` without duplicates. The policies that are not selected as the elites and winners are denoted as discarders `Dis`. After sorting the categories, we perform the crossover operator. Specifically, we obtain the parents $W_{p_1}$ and $W_{p_2}$ by cloning the randomly sampled policies from `E` and `Win` respectively. For each action dimension $a_i$, the weights and biases corresponding to the $a_i$-th action of $W_{p_1}$ will be replaced by those of $W_{p_2}$ with probability $50\%$. Conversely, the weights and biases of $W_{p_2}$ will be replaced by those of $W_{p_1}$. After the exchange, we get two offspring and replace the policies in `Dis` with these offspring. We repeat the above crossover process until all the policies in `Dis` are replaced. Then we perform mutation on all policies except the elites with a probability $90\%$. For the selected agent, we perturb the weights corresponding to each action with probability $\alpha$. For each selected action, the same kind of perturbation is added to a certain percentage (i.e., $\beta$) parameters. Specifically, we perform a small Gaussian perturbation, a large Gaussian perturbation, or reset these parameters with probability $90\%$, $5\%$, and $5\%$, respectively. By completing the above operation, one iteration of population evolution is completed. Except for $\alpha$ and $\beta$ we introduced, we use the same settings of the hyperparameters as in PDERL.

## D  COMPLETE EXPERIMENTAL DETAILS AND FULL RESULTS

This session provides more experiments to help understand ERL-Re$^2$ more comprehensively. We report the average with $95\%$ confidence region based on 5 different seeds. A navigation summary of additional experiments is provided as follows:

**Exp 1**  Ablation study on behavior-level crossover and mutation operators.

**Exp 2**  Evaluation of different choices of $H$ used in the calculation of our surrogate fitness in Eq. 3.

**Exp 3**  Analysis of the repeated episodes used in the calculation of our surrogate fitness in Eq. 3.

**Exp 4**  Evaluation of different choices of the mutation ratio $\beta$ in Algorithm 2.

**Exp 5**  Evaluation of different choices of population size.

**Exp 6**  Experiments on using the idea of ERL-Re$^2$ to improve EA, i.e., Improve EA with the shared state representation and individual policy representations.

**Exp 7**  A comparative experiment on participation rates of the elites and discarders.

**Exp 8**  Experiments on the change of the average/maximum/minimum return of the population before and after optimizing the shared state representation.

**Exp 9**  Further exploration on unsupervised learning to improve shared state representation.

**Exp 10**  Further exploration on the Value Improvement Path (VIP) (Dabney et al., 2021) to improve shared state representation..

---

**Algorithm 2:** Genetic Algorithm in ERL-Re$^2$

---

1 **Input:** the EA population size $n$, the probability $\alpha$ and $\beta$
2 **Initialize:** the shared state representation function $Z_\phi$, the EA population $\mathbb{P} = \{W_1, \cdots, W_n\}$,
3 **repeat**
4    # Obtain Fitness
5    Rollout the EA policies with $Z_\phi$ and get the (surrogate) fitness
6    # Perform selection operators
7    Rank the populations based on fitness and select the best several (1 in ERL-Re$^2$) policies as the elites `E`
8    Randomly select 3 policies other than the elites and retain the best policies as the winner, and repeat the process to collect a set of winners as the winners `Win`
9    Policies that are not selected as the elites and winners are collected as discarders `Dis`.
10    # Perform crossover operators $b$-`Crossover`$(W_{w_1}, W_{w_2})$
11    **while** `Dis` *is not empty* **do**
12       select two discarders $W_{d_i}, W_{d_j}$ from `Dis`
13       random select $W_i$ from `E` and clone $W_i$ as one parent $W_{p_1}$
14       random select $W_j$ from `Win` and clone $W_j$ as the other parent $W_{p_2}$
15       **for** *index $a_i$ in action dimension* **do**
16          **if** *random number < 0.5* **then**
17             $W_{p_1} = W_{p_1} \otimes_{a_i} W_{p_2}$
18          **else**
19             $W_{p_2} = W_{p_2} \otimes_{a_i} W_{p_1}$
20       use $W_{p_1}$ and $W_{p_2}$ to replace $W_{d_i}$ and $W_{d_j}$ and remove $W_{d_i}$ and $W_{d_j}$ from `Dis`
21    # Perform mutation operators $b$-`Mutation`$(W_{p_i})$
22    **for** $W$ *in all non-elite agents* **do**
23       **if** *random number <0.9* **then**
24          **for** *index $a_i$ in action dimension* **do**
25             **if** *random number < $\alpha$* **then**
26                Add minor (90%), drastic (5%) Gaussian perturbations, or reset parameters (5%) to randomly selected $\beta$ parameters from $W_{[a_i]}$
27             **else**
28       **else**
29 **until** *reaching maximum training steps*;

---

**Exp 11** Evaluation of different architectures, i.e., maintaining a PeVFA and an RL Critic for EA and RL separately or only a PeVFA for both EA and RL.

**Exp 12** Time-consuming analysis of ERL-Re$^2$ (TD3) and other related algorithms.

**Exp 13** Analysis of the number of parameters of ERL-Re$^2$ (TD3) and other related algorithms.

**Exp 14** Experiments on the influence of other factors i.e., layer normalization, discount reward, and actor structure.

**Exp 15** More experiments on Deepmind Control Suite (Tassa et al., 2018) with pixel-level inputs.

**Exp 16** More experiments on Starcraft II (Samvelyan et al., 2019).

**Exp 17** The experiments on MuJoCo with longer timesteps.

**Exp 18** Combine ERL-Re$^2$ with SAC and evaluate on MuJoCo.

## D.1 ADDITIONAL ABLATION & HYPERPARAMETER ANALYSIS

In this subsection, we provide experiments on ablation and hyperparameter analysis in **Exp 1**, **Exp 2**, **Exp 3**, **Exp 4**, **Exp 5**, **Exp 6**.

**Exp 1**: To demonstrate the respective roles of operators, we perform ablation experiments on the behavior-level crossover and mutation operators. ERL-Re$^2$ (TD3) w/o crossover(mutation) means using parameter-level crossover(mutation) in ERL to replace the behavior-level crossover(mutation). The results in Fig. 10 show that both behavior-level mutation and crossover operators can deliver performance gains and the best performance can be obtained by using a combination of both.

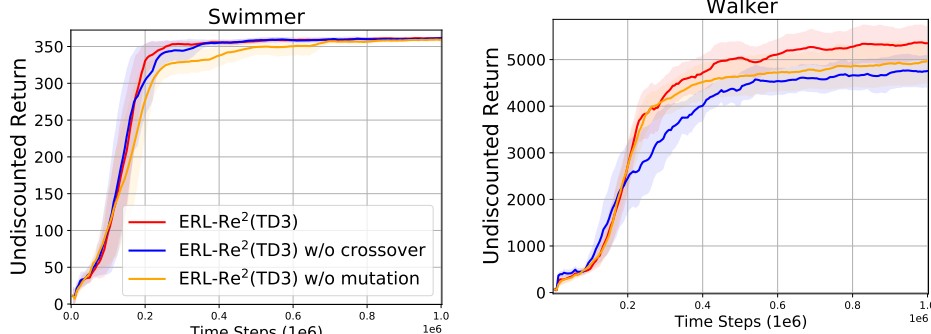

Figure 10: Ablation study on behavior-level crossover and mutation operators in ERL-Re$^2$ (TD3). The results show that both the mutation and crossover can bring performance gains and the best performance can be obtained by using a combination of both.

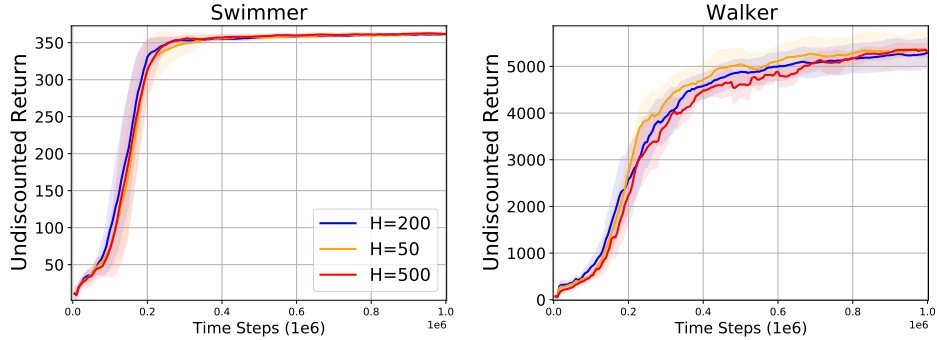

Figure 11: Parameter analysis on ERL-Re$^2$ (TD3) with different $H$. The results show that ERL-Re$^2$ (TD3) is not sensitive to $H$ but proper adjustment can get better results.

**Exp 2**: To analyze the effect of hyperparameter $H$ on performance, we evaluate ERL-Re$^2$ (TD3) with different $H$ on *Swimmer* and *Walker*. The results in Fig. 11 show that ERL-Re$^2$ is not sensitive to $H$, but appropriate adjustments can deliver some performance gains.

**Exp 3**: In the surrogate function $\hat{f}(W_i)$ in Eq. 3, we do not consider the repeated episodes. Here we provide experiments to perform the analysis. The results in Fig.12 show that one episode is often enough to achieve a good performance. To reduce hyperparameters and simplify the tuning process, we set the episode to 1 in all tasks.

**Exp 4**: To analyze the effect of hyperparameter $\beta$ on performance, we evaluate ERL-Re$^2$ (TD3) with different $\beta$ on *Swimmer* and *Walker*. The results in Fig. 13 show ERL-Re$^2$ is not very sensitive to $\beta$. The $\beta$ determines the degree of mutation for an action. A large $\beta$ means strong exploration. According to our experience, in some stable environments such as *Swimmer* and *HalfCheetach*, a larger $\beta$ is generally chosen, while in some unstable environments such as *Walker* and *Hopper*, a smaller $\beta$ is chosen.

**Exp 5**: Although the population size remains consistent 5 across all tasks in our experiments, we give some additional experiments to explore the effect of population size on performance. The results in Fig. 14 show that 5 is more effective than 10 in ERL-Re$^2$ (TD3). The main reason behind this may come from the fact that when the population is too large, the number of steps that the RL agent interacts with the environment is drastically reduced, which may lead to a decrease in RL performance. This problem can be solved by considering a more efficient mechanism for evaluating the individuals to reduce the total interaction of EA properly to improve the number of steps of RL interactions. For a fair comparison, we also evaluate other algorithms based on different population size 5 and 10 and report the best performance.

**Exp 6**: We provide an ablation study on ERL-Re$^2$ without the RL side, which can verify whether the idea of ERL-Re$^2$ can improve EA. Specifically, we construct the policies with the share state representation and individual policy representations in EA and maintain a PeVFA to share the useful

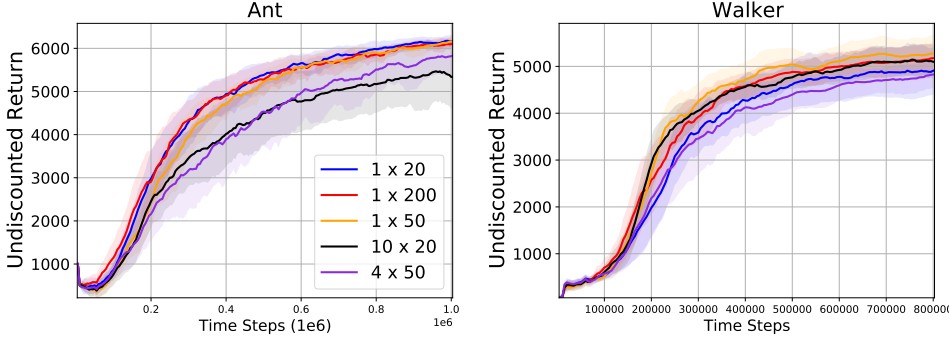

Figure 12: Experiments on the repeated episodes with different $H$ as the surrogate of the fitness. $10 \times 20$ means that repeat 10 episodes with 20 steps (i.e., $H = 20$) and get the average as the fitness. In the results, $1 \times H$ is effective in the tasks. To reduce hyperparameters and simplify the tuning process, we use 1 episode in all tasks.

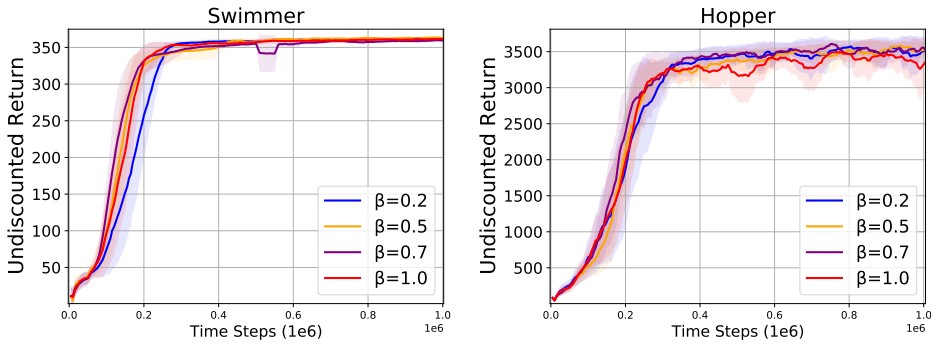

Figure 13: Parameter analysis of the mutation ratio $\beta$ in ERL-Re$^2$ (TD3). The results show that ERL-Re$^2$ (TD3) is not sensitive to $\beta$ but proper adjustment can get better results..

common knowledge in the population. The results in Fig. 15 show that the idea of ERL-Re$^2$ can significantly improve EA, which demonstrates the validity of the idea of ERL-Re$^2$.

### D.2  FURTHER UNDERSTANDING ERL-RE$^2$

In this subsection, we provide experiments to help readers better understand the advantages of ERL-Re$^2$ in **Exp 7** and **Exp 8**.

**Exp 7**: To further investigate the reason why ERL-Re$^2$ is more efficient than ERL and PDERL, we provide a comparative experiment on elites and discarded rates of RL agents. The results in Fig. 16 show that RL agent in ERL-Re$^2$ (TD3) is selected as elites with a larger probability and as discarders with a smaller probability than RL agent in ERL(TD3) and PDERL(TD3), which indicates that both EA and RL in ERL-Re$^2$ (TD3) have more competitive effects on population evolution than in ERL(TD3) and PDERL(TD3).

**Exp 8**: To further study the effect of the state representation on the population, we record the added undiscounted return after optimizing the shared state representation. Specifically, we record the change in average return, maximum return, and minimum return of the population before and after the optimization of the shared state representation. The results in Fig. 17 show the improvement of shared state representations can improve the average reward of the population, where the major improvement comes from poor policies. This demonstrates that the optimization of shared state representations can construct a better policy space for all policies, thus facilitating the evolution of the population.

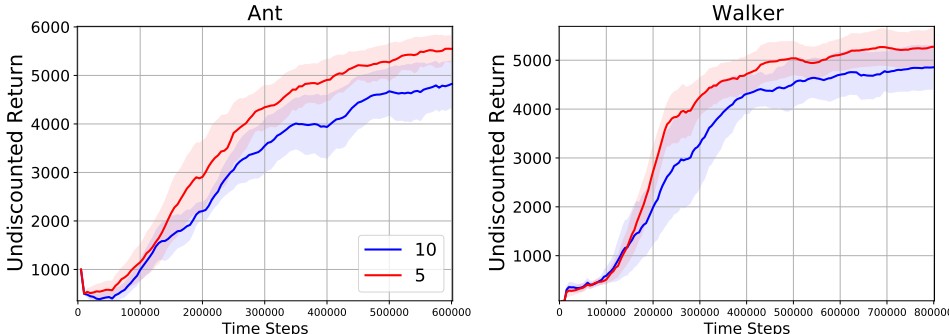

Figure 14: Comparison of ERL-Re$^2$ (TD3) with different population sizes. 5 is more effective than 10 in ERL-Re$^2$ (TD3). Thus we set 5 for population size in all tasks. For a fair comparison, we also evaluate other baselines with 5 and 10 and report the best performance.

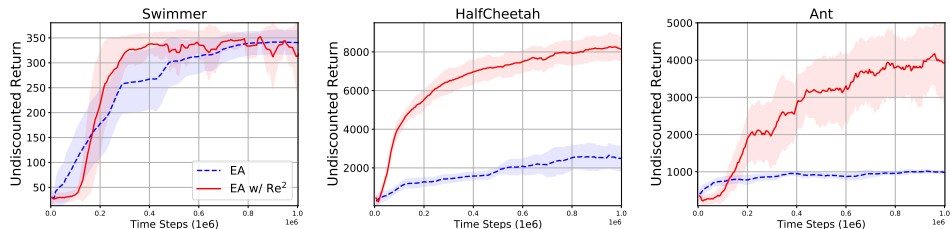

Figure 15: Comparison of EA and EA with Re$^2$. With Re$^2$, the performance of EA can be further improved, which indicates that the idea of ERL-Re$^2$ can be integrated with EA and further improve EA.

### D.3 ADVANCED STATE REPRESENTATION

In this subsection, we further explore how to learn more effective shared state representation in **Exp 9** and **Exp 10**.

**Exp 9**: We conduct additional exploration to enable further improvements of shared state representations by unsupervised learning. To make the representation have dynamic information about the environment, we add the mutual information (MI) loss to the first layer of the shared state representation network. Specifically, When given current state $s$ and the next state $s'$, we want to make the representation based on $s$ and $s'$ have a high correlation with the corresponding action $a$, which can be donated as $I(s, s'; a)$. We use the MINE (Belghazi et al., 2018) to estimate MI. MINE is a neural network to estimate the MI of any two random variables. To make the shared state representation contain the information, we obtain one variable by combining the representations of $s$ and $s'$ from the first layer of the shared state representation network and use the representations and $a$ as the inputs to MINE. We evaluate ERL-Re$^2$ (TD3) with the MI loss and ERL-Re$^2$ on *Ant* and *Walker*. The results in Fig.18 show that the MI loss can bring some performance improvement in *Ant* and fail in *Walker*. We find that the MI loss has both good and bad effects on the final performance. A more stable and efficient way to improve the performance is left as a subsequent work.

**Exp 10**: Inspired by the idea of VIP (Dabney et al., 2021), we conduct experiments to optimize the shared state representation with the previous policy representations. Specifically, we optimize the shared state representation with policy representations not only current but also the previous generation. The results in Fig. 19 show that this approach does not guarantee performance gains. This is just an initial attempt to see how the shared state representation can be further improved by self-supervised learning, which deserves subsequent study. We consider this as future work.

### D.4 DISCUSSION OF OTHER ASPECTS

In this subsection, we further discuss some other aspects of ERL-Re$^2$ such as the architectures selection, time consumption, the number of parameters and other factors in **Exp 11**, **Exp 12**, **Exp 13**, **Exp 14**.

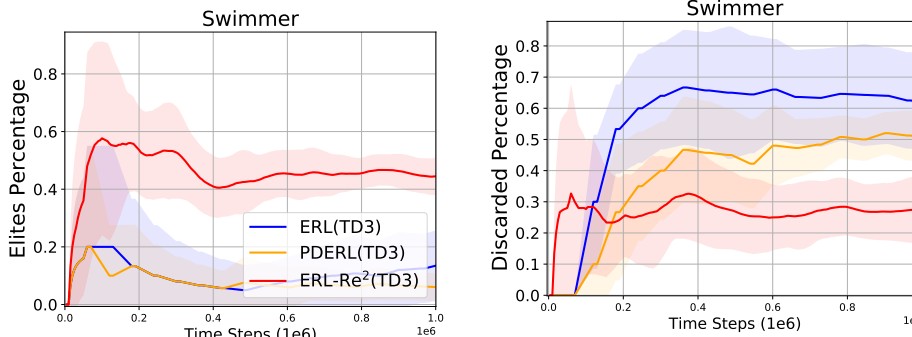

Figure 16: Comparisons of probabilities of RL agent being selected as the elite (left) and discarder (right) of the population in ERL-Re$^2$ (TD3), ERL(TD3), and PDERL(TD3). The results show that both EA and RL in ERL-Re$^2$ (TD3) have more competitive effects on population evolution than in ERL(TD3) and PDERL(TD3).

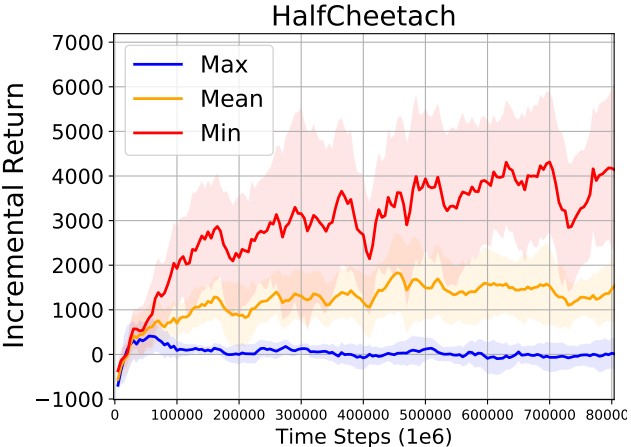

Figure 17: the policy performance change in the EA population after optimizing the shared state representation. Max/Min/Mean means the change of the best/worst/average performance in the population before and after the optimization. The results demonstrate that the optimization of shared state representations can construct a better policy space for all policies, thus facilitating the evolution of the population.

**Exp 11**: In ERL-Re$^2$, we introduce a PeVFA for EA while maintaining the RL critic for RL. In principle, RL can also be improved by the PeVFA. leading to only one PeVFA for both the EA and RL agents. Both architectures can achieve useful common knowledge sharing. To further verify the effectiveness, we conduct comparison experiments between these two architectures in the TD3 version. The results in Fig. 20 show that similar performance can be obtained with both architectures. But to preserve the flexibility and corresponding characteristics of RL methods, we retain the original critic. From the perspective of RL, our architecture does not make changes to the original structure, e.g., maintaining the architecture of the value function, and this separately maintained approach is more convenient to analyze and combine. Therefore, we choose this architecture as the final solution.

**Exp 12**: We evaluate the time consumption of different algorithms on *Walker*. The experiment is carried out on NVIDIA GTX 2080 Ti GPU with Intel(R) Xeon(R) CPU E5-2680 v4 @ 2.40GHz. We evaluate the time consumption of each algorithm individually with no additional programs running on the device. As shown in Table 1, ERL-Re$^2$ incurs some additional time consumption compared to other methods. The main time consumption comes from the training of PeVFA and the shared state representation. $94.73\%$ of the total time is spent on training. If the researchers are sensitive to time overhead, a distributed approach can be used to reduce the time overhead.

**Exp 13**: We provide the number of parameters (i.e., weights and biases) on *HalfCheetach* in Table 2. The results show that ERL-Re$^2$ (TD3) is competitive with other methods in terms of the number

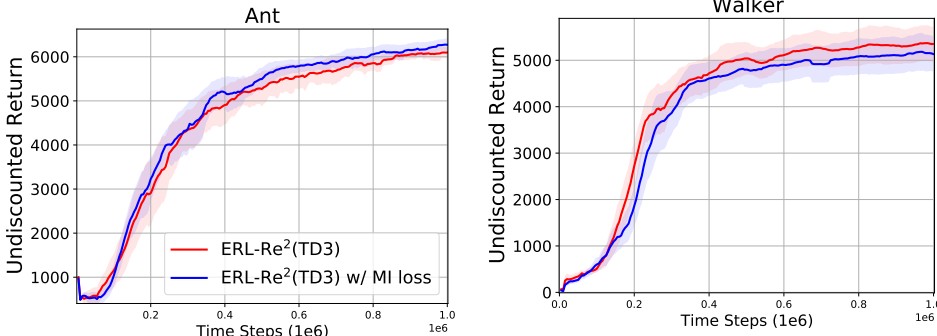

Figure 18: Comparisons of ERL-Re$^2$ (TD3) and ERL-Re$^2$ (TD3) with MI loss (i.e., Self-supervised loss). MI loss brings certain performance improvements in *Ant*, which is worth further research in the future.

of parameters. The number of parameters of ERL-Re$^2$ (TD3) is mainly derived from PeVFA, i.e., $47.9\%$. In addition, ERL-Re$^2$ has better scalability capability than other ERL-related methods as the population size increases. Since ERL-Re$^2$ only introduces a linear policy instead of an independent nonlinear policy network for each agent. For example, on *HalfCheetach*, when the population size increase to 100, ERL-Re$^2$ only need additional $22.4\%$ parameters (i.e., 171570), while other methods such as CERL(TD3) and CEM-TD3 needs additional 1047378600 parameters, which is not feasible.

**Exp 14**: We provide more experiments to eliminate the influence of some other factors. In the official codebase of ERL and PDERL, layer normalization is used for both actor and critic. Our code is built on the codebase of PDERL, thus ERL-Re$^2$ also uses layer normalization. But in the official codebase of TD3, layer normalization is not employed. To exclude the effect of layer normalization, we provide experiments of TD3 with layer normalization. The results in Fig.21 show that TD3 with layer normalization has a similar performance to TD3, which demonstrates that the strengths of ERL-Re$^2$ do not come from the layer normalization.

Second, we want to exclude the influence of discount rewards on performance. Thus we provide experiments by comparing ERL-Re$^2$ (TD3) using MC return of one episode as the surrogate of the fitness and ERL-Re$^2$ (TD3) using discount MC return of one episode as a surrogate of the fitness. The results in Fig. 22 show that only using discounted MC return as the surrogate of the fitness is not effective as using undiscounted MC return, which indicates that the advantages of ERL-Re$^2$ do not come from the discounted form in the surrogate fitness $\hat{f}(W_i)$.

Third, the network structure of the actor of ERL and PDERL is not consistent with the structure of the TD3 paper. Thus we provide a comprehensive comparison, one maintains the structure of the original paper and one is consistent with the structure in the TD3 paper. The results in Fig. 23 and Fig. 24 show that the algorithms with the structure of the TD3 paper improve the performance in some environments and degrade in others. We can find that a large network structure (i.e., the structure of TD3 paper) leads to large fluctuations in ERL such as *Swimmer*, where EA plays a more critical role than RL. The main reason behind this is that parameter-level crossover and mutation in large networks are more unstable than in small networks. In summary, ERL-Re$^2$ (TD3) is still significantly better than the best performance in both structures.

Table 1: Time consumption of different algorithms on *Walker2d* every 10000 steps. The additional time consuming comes mainly from the training of PeVFA and shared state representations. Distributed training can solve this problem.

| Algorithm | ERL-Re$^2$ (TD3) | TD3 | PDERL(TD3) |
|---|---|---|---|
| seconds | 396.14 | 215.5 | 251.37 |
| algorithm | ERL(TD3) | CERL(TD3) | CEM-TD3 |
| seconds | 223.86 | 502.45 | 241.82 |

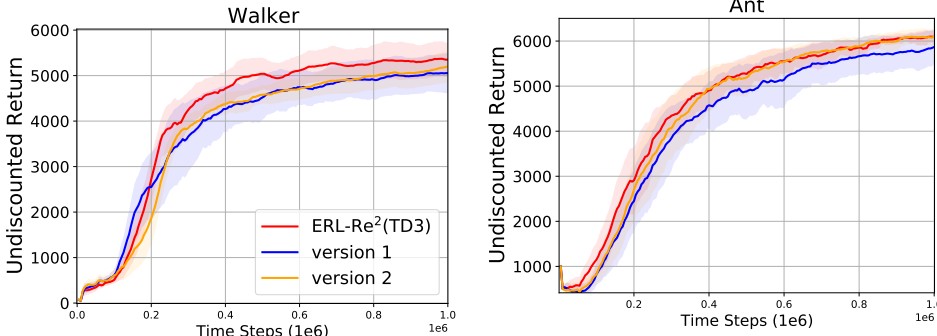

Figure 19: Comparisons of ERL-Re$^2$ (TD3) and ERL-Re$^2$ (TD3) with VIP (Dabney et al., 2021). We design two versions. The version 1 means firstly optimizing PeVFA with the current and previous generation' policy representations, then optimizing the state representations with these policy representations. The version 2 means directly optimizing the state representations with the current and previous generation' policy representations.

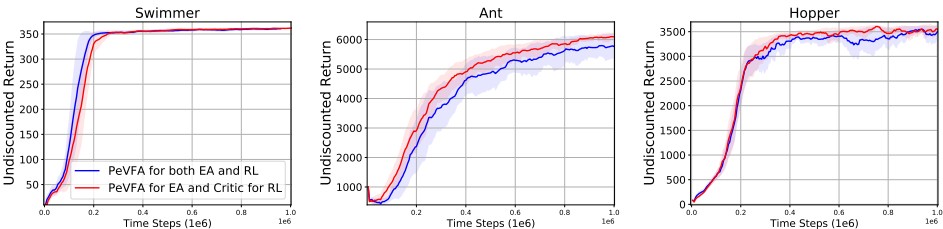

Figure 20: The comparative experiment of two architectures. Only PeVFA for both EA and RL is using the PeVFA to update RL without introducing the RL Critic. PeVFA for EA and Critic for RL is the architecture chosen in this paper. The results show that both architectures are feasible and can achieve competitive performances.

### D.5 EXPERIMENTS WITH VISUAL STATE INPUTS

We evaluate whether ERL-Re$^2$ could bring improvement in Deepmind Control Suite (Tassa et al., 2018) with visual state inputs in **Exp 15**.

**Exp 15**: We evaluate ERL-Re$^2$ on four tasks. In these tasks, we take pixel-level images as inputs. Specifically, we implement ERL-Re$^2$ based on RAD (Laskin et al., 2020) (which can be simply viewed as SAC + Image Augmentation for image inputs). We follow the conventional setting and provide the performance comparison in 100k environment steps. We use 4 representative DMC tasks, covering different robot morphology. The results shown in Table 3 are means and stds over 5 seeds.

We can observe that ERL-Re$^2$ also leads to significant improvements in visual control, which demonstrates the effectiveness of ERL-Re$^2$, while other methods fail. The main reason is that population evolution is very inefficient without knowledge sharing with image inputs, since each individual needs to learn common information from images independently, which is more difficult to learn than the state-level tasks. This also indicates that the techniques of addressing visual states (e.g., image augmentation) are orthogonal to our method, which is worthwhile for further development in the future.

### D.6 EXPERIMENTS IN HIGH-DIMENSIONAL COMPLEX TASKS

For complex high-dimensional tasks, we choose the multi-agent the StarCraft II micromanagement (SMAC) benchmark (Samvelyan et al., 2019) as the testbed for its rich environments and high complexity of control. The SMAC benchmark requires learning policies in a large action space. Agents can move in four cardinal directions, stop, take noop (do nothing), or select an enemy to attack at each timestep. Therefore, if there are $n_e$ enemies in the map, the action space for each allied unit contains $n_e + 6$ discrete actions. To solve these high complexity tasks, we need to integrate ERL-Re$^2$ with Multi-Agent Reinforcement Learning (MARL) algorithm. Fortunately, ERL-Re$^2$ can

Table 2: The number of parameters (i.e., weights and biases) for different algorithms on *HalfCheetach*. For ERL and PDERL, we use | to separate the results for two implementations: the left one uses the same structure as in the original paper, and the right one is aligned with the settings in the TD3 paper.

| Algorithm | ERL-Re$^2$ (TD3) | CERL(TD3) | PDERL(TD3) | CEM-TD3 | ERL(TD3) |
|---|---|---|---|---|---|
| number | 765508 | 1553462 | 456302\|1553462 | 1553462 | 456302\|1553462 |

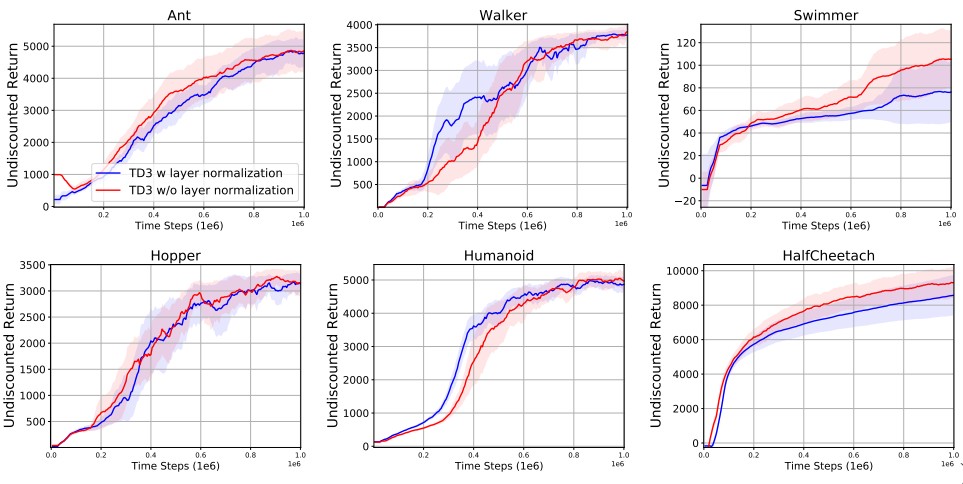

Figure 21: The comparative experiment of TD3 with and without layer normalization. The results demonstrate that the strengths of ERL-Re$^2$ do not come from the layer normalization.

be combined with the MARL algorithms easily. Specifically, we need to maintain a population of teams (i.e., Multiple policies that control multiple agents for collaboration are treated as a team). Each policy in the team needs to follow the two-scale representation-based policy construction. Thus knowledge can be efficiently conveyed through shared representations and reinforcement and evolution occur in the linear policy space.

For the base MARL algorithm, we choose FACMAC, an advanced Multi-Agent Reinforcement Learning algorithm published in NeurIPS 2021 (Peng et al., 2021). Then we combine ERL-Re$^2$ with FACMAC and evaluate it on six tasks. Our implementation is based on the official code of FACMAC[2] and the hyperparameters are kept consistent. To simplify hyperparameter introduced by ERL-Re$^2$, we set $p$ to 1.0, $K$ to 1 on all tasks and select $\beta$ from $\{0.2, 0.05, 0.01\}$. For baselines, we compare ERL-Re$^2$ with MERL (Majumdar et al., 2020) except FACMAC and EA. To the best of our knowledge, MERL is the only work that combines EA and MARL for policy search. The difference is that MERL is applied to environments with dense agent-specific rewards (optimized by RL) as well as sparse team rewards (optimized by EA). For comparison, we use EA and MARL(i.e., FACMAC) to jointly optimize the team rewards of SMAC. The results in Figure 25 show that ERL-Re$^2$ can significantly improve FACMAC in terms of convergence speed and final performance. It is worth noting that the joint action space of *so_many_baneling*, $MMM2$, $3s5z$ and $2c\_vs\_64zg$ is greater than 100 dimensions, which demonstrates that ERL-Re$^2$ can bring improvements to the original algorithm in high-dimensional complex control tasks. Through the above experiments, we further demonstrated the efficiency and generalization of ERL-Re$^2$.

## D.7 EXPERIMENTS ON MUJOCO WITH LONGER TIMESTEPS

In Figure 4, we provide the experiments on MuJoCo with 1 million timesteps, which is mainly consistent with the training timestep in SAC (Haarnoja et al., 2018) and TD3 (Fujimoto et al., 2018). To demonstrate the competitiveness of ERL-Re$^2$ in longer timesteps, we provide the experiments in the TD3 version with 3 million timesteps. Since *Swimmer* and *Hopper* have already achieved convergence performance, we mainly provide the experiments on the remaining four tasks. The results may differ slightly from those in the main text, as they are obtained by running on different

---

[2]https://github.com/oxwhirl/facmac

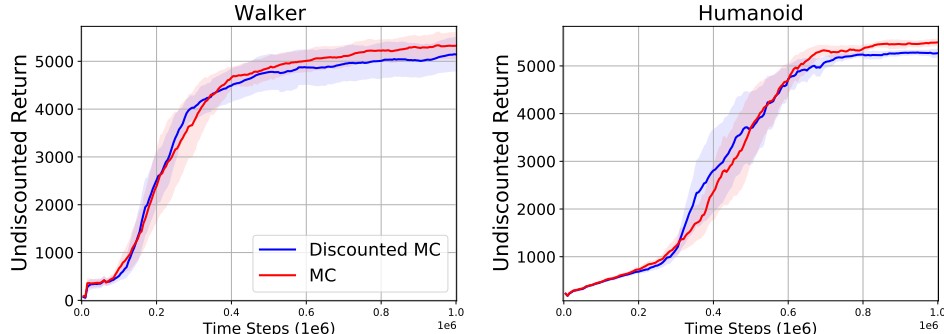

Figure 22: Comparison experiment using one episode MC return and one episode discounted MC return. The experiments do not use $\hat{f}(W_i)$ as a surrogate of the fitness. The results indicate the strengths of ERL-Re$^2$ do not come from the discounted design.

Table 3: Comparisons on Deepmind Control Suite with image inputs.

| Environments | Cartpole Swingup | Cheetah Run | Finger Turn hard | Walker Walk |
|---|---|---|---|---|
| ERL-Re$^2$ (RAD) | $854.453 \pm 17.582$ | $508.619 \pm 22.622$ | $197.35 \pm 83.388$ | $567.417 \pm 139.278$ |
| RAD | $728.66 \pm 149.219$ | $472.594 \pm 29.595$ | $126.725 \pm 48.954$ | $498.867 \pm 142.509$ |
| ERL(RAD) | $527.81 \pm 126.48$ | $329.709 \pm 132.674$ | $93.6 \pm 130.53$ | $287.969 \pm 57.168$ |
| PDERL(RAD) | $471.164 \pm 220.221$ | $331.903 \pm 29.614$ | $150 \pm 70.711$ | $252.492 \pm 93.01$ |
| CEM-RAD | $694.909 \pm 44.74$ | $421.574 \pm 20.891$ | $50.25 \pm 67.67$ | $66.042 \pm 19.23$ |
| Steps | 100k | 100k | 100k | 100k |

servers. The results in Figure 26 show ERL-Re$^2$ still significantly outperforms other algorithms with 3 million timesteps, which further verifies the efficiency of ERL-Re$^2$.

# E  METHOD IMPLEMENTATION DETAILS

All experiments are carried out on NVIDIA GTX 2080 Ti GPU with Intel(R) Xeon(R) CPU E5-2680 v4 @ 2.40GHz.

## E.1  IMPLEMENTATION OF BASELINES

For all baseline algorithms, we use the official implementation. In our paper, there are two main implementations, one combines TD3 and the other combines DDPG. Many of these official implementations contain two implementations such as CEM-RL[3] and CERL[4]. For methods that do not have two implementations such as ERL[5] and PDERL[6] (both based on DDPG), we modify the DDPG to TD3 which is a very simple and straightforward extension. For the basic algorithm TD3[7], we use the official implementation. For EA, we use the official implementation in PDERL. For SAC and PPO, we use the implementation from *stable-baseline3*[8]. We fine-tuned ERL-related baselines, mainly including population size, episodes of agents interacting with the environment in the population, elite rate, injection frequency, etc.

## E.2  NETWORK ARCHITECTURE

This section details the architecture of the networks. Our code is built on the official codebase of PDERL. Most of the structure remains the same. For structures specific to our framework, the shared

---

[3]https://github.com/apourchot/CEM-RL

[4]https://github.com/intelai/cerl

[5]https://github.com/ShawK91/Evolutionary-Reinforcement-Learning

[6]https://github.com/crisbodnar/pderl

[7]https://github.com/sfujim/TD3

[8]https://github.com/DLR-RM/stable-baselines3

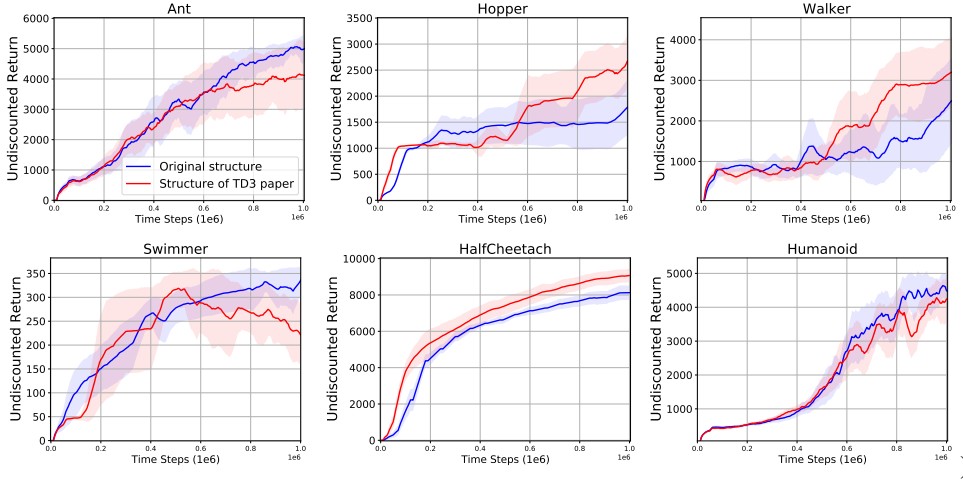

Figure 23: The comparative experiment of ERL with different actor structures. With the structure of the TD3 paper, ERL improves performance in some tasks and degrades in other tasks. The parameter-level operators in a large network can make performance easily crash. (See *Swimmer*). In summary, ERL-Re$^2$ (TD3) is still significantly better than the best performance of the two structures.

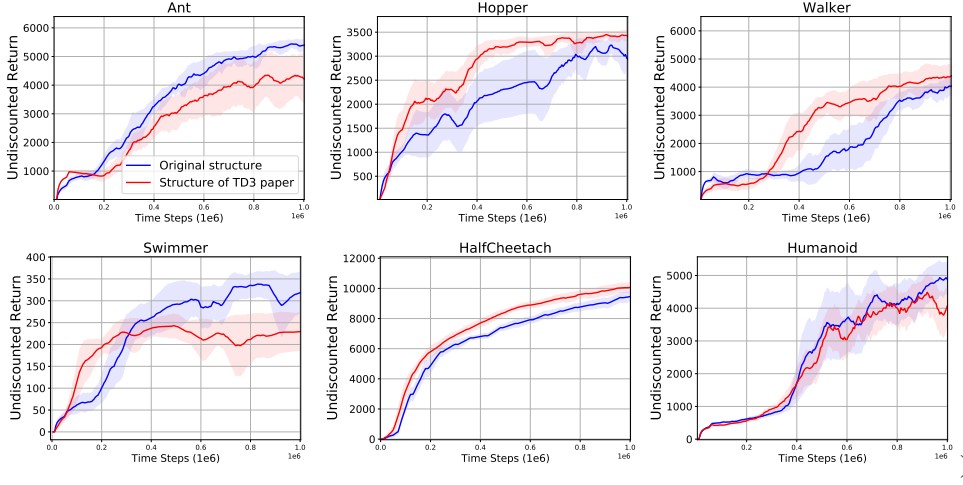

Figure 24: The comparative experiment of PDERL with different actor structures. With the structure of the TD3 paper, PDERL improves performance in some tasks and degrades in other tasks. In summary, ERL-Re$^2$ (TD3) is still significantly better than the best performance of the two structures.

state representation network is constructed by two fully connected layers with 400 and 300 units. The policy representation is constructed by one fully connected layer with `action_dim` units.

PeVFA takes state, action and policy representation as inputs and maintains double $Q$ networks which are similar to TD3. The policy representation can be regarded as a combination of a matrix with shape $[300, \texttt{action\_dim}]$ (i.e., weights) and a vector with shape $[\texttt{action\_dim}]$ (i.e., biases) which can be concatenated as a matrix with shape $[300 + 1, \texttt{action\_dim}]$. We first encode each vector with shape $[300 + 1]$ of the policy representations with 3 fully connected layers with units 64 and `leaky_relu` activation function. Thus we can get an embedding list with shape $[64, \texttt{action\_dim}]$ and get the final policy embedding with shape $[64]$ by taking the mean value of the embedding list in the action dimension. With the policy embedding, we concatenate the policy embedding, states, and actions as the input to an MLP with 2 fully connected layers with units 400 and 300 and get the predicted value by PeVFA. The activation functions in PeVFA all use `leaky_relu`. We list structures in Table 4 and 5. For other structures, we take the settings directly from the codebase of PDERL. The network structure is the same in ERL-Re$^2$ (TD3) and ERL-Re$^2$ (DDPG).

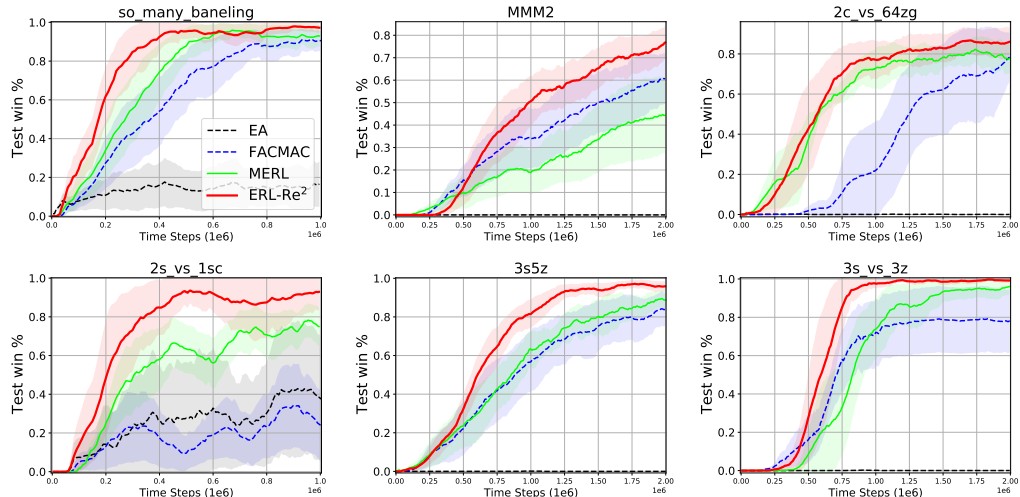

Figure 25: Performance comparison between ERL-Re$^2$ and baselines (all in the FACMAC version).

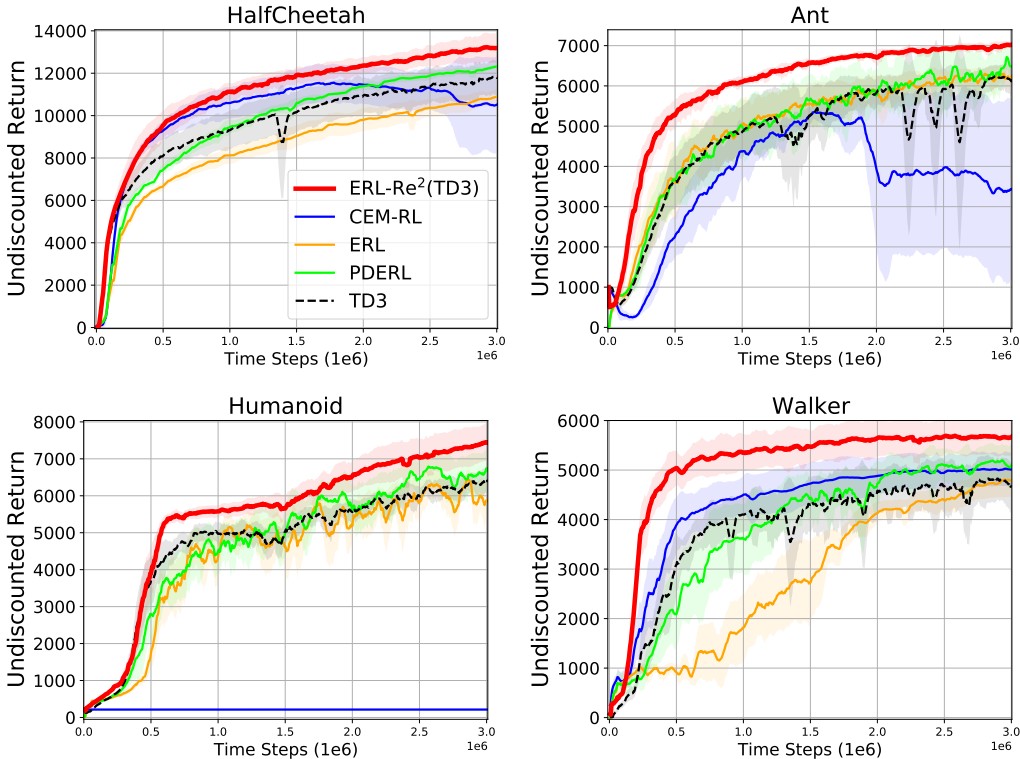

Figure 26: Performance comparison between ERL-Re$^2$ and baselines (all in the TD3 version with 3 million timesteps).

### E.3 HYPERPARAMETERS

This section details the hyperparameters across different tasks. Some hyperparameters are kept consistent across all tasks. Population size is 5 for both ERL-Re$^2$ (TD3) and ERL-Re$^2$ (DDPG). $\alpha$ for mutation operators is 1.0 for both ERL-Re$^2$ (TD3) and ERL-Re$^2$ (DDPG). The number of episodes of interaction with the environment for surrogate fitness is 1. $\gamma$ is 0.99 except 0.999 for Swimmer (all baselines follow this setting). Since ERL-Re$^2$ is built on PDERL, other settings remain

Table 4: The structures of the shared state representation network and policy representations.

| Shared State Representation Network | Policy Representation |
|---|---|
| (`state_dim`, 400)
tanh
(400,300)
tanh | (300, `action_dim`)
tanh |

Table 5: The structure of PeVFA.

| PeVFA | |
|---|---|
| (`state-action_dim` + 64, 400)
`leaky_relu`
(400, 300)
`leaky_relu`
(300, 1) | (301, 64)
`leaky_relu`
(64, 64)
`leaky_relu`
(64, 64) |

the same. In addition, we list other hyperparameters specific to ERL-Re$^2$ which varied across tasks in Table 6 and Table 7. In addition to this, we find that ERL-Re$^2$ is very stable, and **if you don't want to tune too many hyperparameters**, competitive performance can be obtained when keeping $K$ and $\beta$ consistent with the settings in the table and do not use our surrogate fitness (i.e., $p$=1.0). We prove this in Fig.6d.

Table 6: Details of the hyperparameters of ERL-Re$^2$-TD3 that are varied across tasks.

| Env name | $p$ | $\beta$ | $H$ | $K$ |
|---|---|---|---|---|
| HalfCheetach | 0.3 | 1.0 | 200 | 1 |
| Walker | 0.8 | 0.2 | 50 | 1 |
| Swimmer | 0.3 | 1.0 | 200 | 3 |
| Hopper | 0.8 | 0.2 | 50 | 3 |
| Ant | 0.5 | 0.7 | 200 | 1 |
| Humanoid | 0.5 | 0.5 | 200 | 1 |

## F ADDITIONAL DISCUSSION

### F.1 THE ALTERNATIVE OF REPLACING GA BY CEM

In this paper, we use GA as the basic evolutionary algorithm to combine with different RL algorithms. ERL-Re$^2$ can utilize different evolutionary algorithms to exploit the advantages of these algorithms. Here we discuss how to use CEM (Pourchot & Sigaud, 2019) to replace GA in ERL-Re$^2$.

The pseudo-code is shown in Algorithm 3. To achieve this, we need to modify two aspects: 1) population generation. 2) population evolution. For the first aspect, we need to use CEM to generate the population instead of pre-defining the entire population by constructing a mean policy $W_\mu$ to sample the population based on a predefined covariance matrix $\Sigma = \sigma_{init}\mathcal{I}$. For the second aspect, we need to select the top $T$ policies to update $W_\mu$ and $\Sigma = \sigma_{init}\mathcal{I}$ instead of the crossover and mutation operators.

### F.2 TRAINING PEVFA IN AN OFF-POLICY FASHION

To our knowledge, PeVFA (Tang et al., 2022) and related variants are trained in an on-policy fashion in prior works. In the original paper of PeVFA, a newly implemented PPO algorithm, called PPO-PeVFA is studied. In PPO-PeVFA, the algorithm saves states, returns and the corresponding policies in a replay buffer; then a PeVFA $\mathbb{V}_\theta(s, \chi_\pi)$ is trained by the Monte-Carlo method, based on the experiences collected in the replay buffer with some policy representation $\chi_\pi$. Note that although historical experiences are replayed, each policy uses its own collected experiences to train $\mathbb{V}_\theta$, thus being on-policy. Training the PeVFA $\mathbb{V}_\theta$ in off-policy manner needs to introduce additional mechanisms such as importance sampling method, which are not implemented in their work. Similarly, this case can be also be found in (Faccio et al., 2021).

Table 7: Details of the hyperparameters of ERL-Re$^2$-DDPG that are varied across tasks.

| Env name | $p$ | $\beta$ | $H$ | $K$ |
|---|---|---|---|---|
| HalfCheetach | 0.5 | 1.0 | 200 | 1 |
| Walker | 0.8 | 0.2 | 50 | 1 |
| Swimmer | 0.3 | 0.5 | 200 | 3 |
| Hopper | 0.8 | 0.7 | 50 | 3 |
| Ant | 0.7 | 0.5 | 200 | 1 |
| Humanoid | 0.7 | 0.5 | 200 | 1 |

---

**Algorithm 3:** ERL-Re$^2$ with Cross Entropy Method (CEM)

---

1 **Input:** $\Sigma = \sigma_{init}\mathcal{I}$ and $T$ of CEM, the EA population size $n$, the probability $p$ of using MC estimate, the partial rollout length $H$

2 **Initialize:** Initialize the mean policy $W_\mu$ of CEM. a replay buffer $\mathcal{D}$, the shared state representation function $Z_\phi$, the RL agent $W_{rl}$, the RL critic $Q_\psi$ and the PeVFA $\mathbb{Q}_\theta$ (target networks are omitted here)

3 **repeat**

4      # Draw the CEM population $\mathbb{P}$

5      Draw the population $\mathbb{P} = \{W_1, \cdots, W_n\}$ from $\mathcal{N}(\pi_\mu, \Sigma)$

6      # Rollout the EA and RL agents with $Z_\phi$ and estimate the (surrogate) fitness

7      **if** *Random Number > p* **then**

8          Rollout each agent in $\mathbb{P}$ for $H$ steps and evaluate its fitness by the surrogate $\hat{f}(W)$    ▷ see Eq. 3

9      **else**

10          Rollout each agent in $\mathbb{P}$ for one episode and evaluate its fitness by MC estimate

11      Rollout the RL agent for one episode

12      Store the experiences generated by $\mathbb{P}$ and $W_{rl}$ to $\mathcal{D}$

13      # *Individual* scale: evolution and reinforcement in the policy space

14      Train PeVFA $\mathbb{Q}_\theta$ and RL critic $Q_\psi$ with $D$    ▷ see Eq. 1

15      **Optimize CEM:** rank all policies with fitness and use the top $T$ policies to update CEM.

16      **Optimize the RL agent:** update $W_{rl}$ (by e.g., DDPG and TD3) according to $Q_\psi$    ▷ see Eq. 5

17      # *Common* scale: improving the policy space through optimizing $Z_\phi$

18      **Update the shared state representation:** optimize $Z_\phi$ with a unifying gradient direction derived from value function maximization regarding $\mathbb{Q}_\theta$ and $Q_\psi$    ▷ see Eq. 2

19 **until** *reaching maximum training steps*;

---

In this work, we train PeVFA in an off-policy manner for the first time. We achieve this in a more efficient way by training a PeVFA $\mathbb{Q}_\theta(s, a, W)$ with 1-step TD learning and the linear policy representation $W$. That is, the value function of each policy can be learned (with its representation $W_i$ and $\mathbb{Q}_\theta$) from any off-policy experience $(s, a, s', r)$ according to Eq. 1. We emphasize that this is appealing, and we believe that this can realize effective training of PeVFA. In turn, the ability of PeVFA in conveying the value generalization among policies can be better rendered. We will further investigate the potential of this point in the future.

We use PeVFA to provide value estimates for each agent (in the EA population). One may note that the same purpose can be achieved by maintaining a conventional $Q$ network for each agent individually. However, this means suffers from the issue of scalability, resulting in significant time and resource overhead. In addition, this approach does not use policy representations and thus cannot take advantage of value function generalization to improve value function learning.

