# OpenReview forum: "ERL-Re$^2$: Efficient Evolutionary Reinforcement Learning with Shared State Representation and Individual Policy Representation "
_ICLR.cc/2023/Conference — ICLR 2023 poster_

### Official Review · Reviewer_ouqq · 2022-10-25

**Confidence:** 4
**Correctness:** 3
**Technical Novelty And Significance:** 3
**Empirical Novelty And Significance:** 3
**Recommendation:** 8

**Clarity, Quality, Novelty And Reproducibility:**

As mentioned in the above section, this paper is generally clearly written. The approach is novel. The algorithmic and experimental details are provided.

**Strength And Weaknesses:**

# Strength

* Strong performance has been observed on MuJoCo environments in comparison with PPO, SAC, DDPG, TD3 (RL baselines), ERL, CERL, PDERL, CEM-RL (ERL baselines).
* Overall, the paper is well-written. The algorithm and experimental details are provided sufficiently.

# Weaknesses

* From the description of the algorithm written in the main text, the introduction of the shared state representation appears to be aimed at improving search efficiency. However in the experiments, the final performance was improved on some environments. Is this because the training was not finished? A possible criticism is that as the training is not finished, the final performance of the proposed approach might be lower than the other approaches. A careful discussion is required.

# Comment / Question

* The proposed behavior-level operators seem to make sense. However, similar effect (considering the correlation between parameters) could be realized by just using CMA-ES or CEM. Why do you want to employ a GA-like approaches?

**Summary Of The Paper:**

Combinations of evolutionary algorithm (EA) and deep reinforcement learning (DRL) have been investigated in the literature to take advantage of EA (exploration ability, robustness and stability) while maintaining the advantage of DRL (sample efficiency). However, in the previous work on the combination of EA and DRL, we observe that the performance of the combination is sometimes dominated by EA-only or DRL-only approaches, for example on Swimmer and Humanoid on MuJoCo environments.

This paper proposed a novel combination, namely Evolutionary Reinforcement Learning with Two-scale State Representation and Policy Representation (ERL-Re2). In the proposed approach, the policies of EA and RL agents are linear policies on a non-linear state-representation that is shared over all EA and RL agents. The state-representation is trained using a replay buffer storing all the experiences generated by EA and RL. Linear policies are trained independently by EA and RL. To improve the sample efficiency of EA part, the authors introduced a shared critic network taking the policy parameter as its input and use it as a surrogate function of the cumulative reward, leading to reduction of the number of interaction for policy evaluation in EA. Moreover, to make the crossover and mutation in EA part more sense, the authors introduced novel operators, named behavior-level mutation and crossover.

Numerical experiments on MuJoCo environments show its efficiency as well as superior final performance over existing ERL variants and RL baselines and EA.

**Summary Of The Review:**

This paper improved the performance of ERL significantly as summarized in the strength and weaknesses section. Each algorithmic component have been carefully evaluated in experiment (in appendix).

---

> ### Author Response · Authors · 2022-11-14
> **Initial Response to Reviewer ouqq (Part 2/2)**
>
>
> >2. [Re: “The proposed behavior-level operators seem to make sense. However, similar effect (considering the correlation between parameters) could be realized by just using CMA-ES or CEM. Why do you want to employ a GA-like approaches?”]
>
> Our work follows the previous series of works, i.e., ERL (Khadka et al., 2018) and PDERL (Bodnar et al., 2020) and use GA as the algorithm component for the EA side, **to be consistent and to avoid unnecessary distinctions**.
>
> For behavior-level operations (or perturbations), we mean that the effect of the operation (or perturbation) is controllable to change specific action dimension(s) (with no effect on the other ones).
> We think there are **two possible ways to realize behavior-level perturbations with CMA-ES or CEM**:
>
> (1) One simple way is to only perturb the final-layer parameters corresponding to the output of specific action dimension(s), similar to what we do for the linear policy representation in ERL-Re$^2$.
> This is straightforward because for non-final layers, a noise added to a parameter may result in uncontrollable effect on the output of different action dimensions (unless special network architectures are used).
> However, for typical CMA-ES or CEM, it is irrational to only perturb the final layer and to leave other layers unchanged. One possible choice is to implement CMA-ES or CEM under our two-scale representation framework.
>
> (2) Another possible way is to compute the parameter gradient regarding specific action dimension(s) and find the orthogonal gradient direction.
> Then behavior-level perturbation may be realized by adding parameter noises according such orthogonal directions.
>
> We appreciate the reviewer's inspiring question and we will consider them in our future work.
>
> &nbsp;
>
> **Reference:**
>
> [1] Will Dabney, André Barreto, Mark Rowland, Robert Dadashi, John Quan, Marc G. Bellemare, David Silver.
> The Value-Improvement Path: Towards Better Representations for Reinforcement Learning. AAAI 2021
>
> [2] Evgenii Nikishin, Max Schwarzer, Pierluca D'Oro, Pierre-Luc Bacon, Aaron C. Courville.
> The Primacy Bias in Deep Reinforcement Learning. ICML 2022
>
> [3] Clare Lyle, Mark Rowland, Will Dabney.
> Understanding and Preventing Capacity Loss in Reinforcement Learning. ICLR 2022
>
> [4] Aviral Kumar, Rishabh Agarwal, Tengyu Ma, Aaron C. Courville, George Tucker, Sergey Levine.
> DR3: Value-Based Deep Reinforcement Learning Requires Explicit Regularization. ICLR 2022
>
> ------
> We hope our replies have addressed all the concerns the reviewer posed and shown the improved quality of the paper. **We are always willing to answer any of the reviewer's concerns** about our work and we are looking forward to more inspiring discussions.

---

> ### Author Response · Authors · 2022-11-14
> **Initial Response to Reviewer ouqq (Part 1/2)**
>
> We sincerely appreciate the reviewer's careful review and constructive comments, and we would like you to know that your questions provide considerably helpful guidance to improve the quality of our paper.
> We will try our best to address each of the concerns and questions raised by the reviewer below:
>
> >1. [Re: “... the introduction of the shared state representation appears to be aimed at improving search efficiency. However in the experiments, the final performance was improved on some environments. Is this because the training was not finished? A possible criticism is that as the training is not finished, the final performance of the proposed approach might be lower than the other approaches. A careful discussion is required.”]
>
> We answer this question from algorithmic and experimental angles in the following.
>
> For the algorithm angle,
> as we mention in the introduction, the state representation learned by individual agents can be redundant and specialized.
> Therefore, the advantages of shared state representation are two-fold:
> first, it reduces the redundant representation learning (thus be more efficient);
> second, it strengthens the expressivity of representation which allows better policies to be found by EA and RL agents in a better policy space (thus potentially better final performance).
> Several works also discover and study the problematic representation specialization of single (RL) agent learning [1-4].
>
> For the experimental angle, we followed the number of interaction steps (i.e., 1 million timesteps) used by TD3 and SAC for training on OpenAI MuJoCo. ERL-Re$^2$ brings significant improvements to TD3 or DDPG (black dashed line) on all tasks, while other methods fail, which proves that ERL-Re$^2$ has higher sample efficiency and can find better policies.
> To address the reviewer's concerns, **we provide additional results with the longer horizon (i.e., 3 million timesteps) on *HalfCheetah*, *Ant*, *Humanoid* and *Walker*** in the revised version.
> (We have achieved the converged performance on *Hopper* (3500) and *Swimmer* (350), thus these tasks are not considered.)
> The results in Figure 26 show that ERL-Re$^2$ still outperforms other methods with 3 million timesteps, which also demonstrates the ability of ERL-Re$^2$ to maintain high sample efficiency and clear performance advantages over a longer training horizon. See Appendix D.7 for details.

---

> > ### Comment · Reviewer_ouqq · 2022-11-18
> > **Thank you for the clarification**
> >
> > One minor comment: The motivation of the introduction of the architecture (or algorithm) could be better explained already in the introduction, possibly in relation to what you actually evaluate in the experiments.

---

> > > ### Author Response · Authors · 2022-11-22
> > > **Thank you for your valuable comments!**
> > >
> > > Thank you for your valuable comments, we will consider incorporating experimental findings into the introduction and presentation of our framework in subsequent versions to better illustrate our motivation.

---

### Official Review · Reviewer_oDSW · 2022-10-26

**Confidence:** 4
**Correctness:** 4
**Technical Novelty And Significance:** 3
**Empirical Novelty And Significance:** 3
**Recommendation:** 6

**Clarity, Quality, Novelty And Reproducibility:**

The paper is well-written and easy to follow. The idea is novel. I believe that the experiments can be reproduced according to the details in the paper and appendix.

**Strength And Weaknesses:**

- It is a good and reasonable idea to split the policy into the shared state representation and the independent policy representation, which may be of independent interest.

- The method performs significantly better than other methods in the experiments, and the ablation studies are extensive.

- The paper is well-written and easy to follow.

- The method may be further improved by taking full advantage of EAs, e.g., keeping the diversity of population.

Minor problems: There are some confusing symbols in the paper, e.g., the symbols of the sets in Section 6.1.


**Summary Of The Paper:**

This paper points out the problems of the previous ERL frameworks, i.e., the parameters of different policies are independent and the EA operators often fail. The paper then proposes a new ERL framework that splits the policy network into two parts, i.e., the shared nonlinear state representation and the independent linear policy representation, to overcome the former problem, and proposes the behavior-level EA operators to overcome the latter problem. Experimental results show that the proposed method performs better than other ERL, EA, and RL methods in all tasks.

**Summary Of The Review:**

This paper proposes a new ERL framework that splits the policy into two parts, and proposes the behavior-level EA operators. The idea is novel and reasonable. The paper conducts extensive experiments and ablation studies to show the performance of the framework and the effects of each part of the framework. Despite some limitations, it is overall a good work.

---

> ### Author Response · Authors · 2022-11-14
> **Initial Response to Reviewer oDSW**
>
> We sincerely appreciate the reviewer's careful review and constructive comments, and we would like you to know that your questions provide considerably helpful guidance to improve the quality of our paper.
>
> We will try our best to address each of the concerns and questions raised by the reviewer below:
>
>
> >1. [Re: “The method may be further improved by taking full advantage of EAs, e.g., keeping the diversity of population.”]
>
> We agree with the reviewer's opinion. In this work, our focus lies on the new ERL framework established based on two-scale representation, while the potential of both EA and RL sides is not fully explored.
> We take the study on better ways to make EA and RL work together as our main future direction.
> Here, we briefly discuss several worthwhile angles below:
>
> (1) For the EA side,
> it is possible to further improve ERL-Re$^2$ by taking full advantage of more advanced concepts and algorithms in EA community.
> As mentioned by the reviewer, explicitly encouraging or keeping diversity is a potential approach.
> In Appendix A, we presented integrating the principle of Quality Diversity (QD) as an important future direction, which maximizes the performance (quality) while optimizing the diversity of the population.
>
> (2) For the RL side, a promising direction is to use more advanced experience replay algorithms.
> In this work, the RL agent samples uniformly from the experience buffer.
> Since the experience samples are collected by multiple policies, each of which has (almost) an independent policy optimization process,
> the experience can be diverse with different policy behaviors.
> For an ideal experience replay algorithm that makes the best use of beneficial experiences or replays in the best order, learning problems like passive learning [1] and TD error propagation [2,3] may need to be considered.
>
> In addition, we found the potential of EA can be further developed in Multi-Agent RL (MARL).
> In our additional experiments on the StarCraft II micromanagement (SMAC) benchmark (Figure 25 and Appendix D.6 in our revision),
> we found that maintaining a population of teams (each consisting of multiple agents) and performing agent-level GA operations can also achieve significant improvements over both MAEA and MARL baselines in such environments with the high complexity of control.
>
>
> &nbsp;
>
> >2. [Re: Minor problems]
>
> The symbols of the sets in Section 6.1 are amended in our revision.
> We also proofread other places for better correctness.
>
> &nbsp;
>
> **Reference:**
>
> [1] Georg Ostrovski, Pablo Samuel Castro, Will Dabney. The Difficulty of Passive Learning in Deep Reinforcement Learning. NeurIPS 2021
>
> [2] Aviral Kumar, Abhishek Gupta, Sergey Levine. DisCor: Corrective Feedback in Reinforcement Learning via Distribution Correction. NeurIPS 2020
>
>
> [3] Zhang-Wei Hong, Tao Chen, Yen-Chen Lin, Joni Pajarinen, Pulkit Agrawal. Topological Experience Replay. ICLR 2022
>
>
> ------
> **We are always willing to answer any of the reviewer's concerns** about our work and we are looking forward to more inspiring discussions.

---

> > ### Author Response · Authors · 2022-11-30
> > **Further Discussions**
> >
> > Dear Reviewer oDSW,
> >
> > Thank you again for reviewing our paper. As the discussion phase has been going on for some time, we'd like to ask one more time if there are further clarifications/modifications about our paper that you like us to address, or if any of the responses to your questions need further elaboration.
> >
> >
> > We also try our best to make the experiments more comprehensive. Specifically, we have added experiments on SMAC (in Appendix D.6 and Figure 25) and on DMC (in Appendix D.5 and Table 3). The results show that ERL-Re$^2$ significantly improves the basic algorithms (i.e., FACMAC and RAD) and outperforms other baselines, which further demonstrates the effectiveness of ERL-Re$^2$.
> >
> >
> > If you believe that our response and revision have addressed your concerns properly, please re-evaluate the contributions of our paper.
> >
> > Thank you,
> >
> > Paper 931 Authors

---

> > ### Comment · Reviewer_oDSW · 2022-12-06
> > **Thanks for your reply**
> >
> > Thanks for your reply. I have no further comment.

---

### Official Review · Reviewer_vR7r · 2022-11-03

**Confidence:** 3
**Correctness:** 3
**Technical Novelty And Significance:** 3
**Empirical Novelty And Significance:** 3
**Recommendation:** 6

**Clarity, Quality, Novelty And Reproducibility:**

The paper is mostly clear. The writing could be improved in places, but it does not cause confusion.

I do not feel that I am familiar enough with the EA literature to evaluate the novelty of the proposed behavior-level crossover and mutation.

Regarding the fitness evaluation metric: I think “novel” is a strong word, seeing as the $H$-step return estimate has been used in other works such as A3C and Rainbow. Also, it is not accurate to call it $H$-step “TD” as there is no temporal difference involved.

The paper includes hyperparameter settings and references to which specific codebases were used. Therefore, I’m inclined to believe it can be reproduced.

**Strength And Weaknesses:**

Strengths
* The core idea of utilizing a shared representation across agents and only applying EA to the last layer makes intuitive sense.
* On all the tasks used in evaluation, ERL-Re$^2$ outperforms existing RL and EA algorithms.
* The appendix includes extensive ablations investigating the effects of various hyperparameters and design choices. I appreciate that the authors have reported negative results for different ideas they tried; this may save a grad student some time later.

Weaknesses
* Since the algorithm combines elements of RL and EA, it has many hyperparameters to tune.
* The performance gains resulting from the proposed modifications (behavior-level genetic operators and fitness evaluation metric) are not very large.

**Summary Of The Paper:**

The paper proposes a strategy for combining aspects of reinforcement learning (RL) and evolutionary algorithms (EA). A shared representation is learned for all actors (EA policies and an RL policy), with each EA actor maintaining only its own linear head. The EA policies are recombined using proposed behavior-level genetic operators that act on rows of the linear head, while the RL policy is updated by maximizing the Q function as usual. The authors show that EA agent performance can be estimated without full rollouts by using the learned value function. The proposed algorithm, dubbed ERL-Re$^2$, displays strong performance on various control tasks.

**Summary Of The Review:**

Overall, I think the paper is a useful contribution owing to its strong empirical results. The RL community would benefit from seeing that ideas from EA can be effectively combined with RL to improve performance.

---

> ### Author Response · Authors · 2022-11-14
> **Initial Response to Reviewer vR7r (Part 2/2)**
>
>
> >3. [Re: “I think 'novel' is a strong word, seeing as the $H$-step return estimate has been used in other works such as A3C and Rainbow.”]
>
> We are aware of the use of $H$-step bootstrapping return/value estimate in DRL literature.
> We clarify the novelty of the proposed surrogate fitness in the context of ERL in the following.
>
>
> **The major point lies in the correction of bias in fitness** (or expected return) estimation in the previous (actually concurrent) surrogate fitness (Wang et al., 2022).
> This is because, even with $H$-step bootstrapping estimates, bootstrapping from the RL critic as done by (Wang et al., 2022) is biased.
> Thus, the key thing here is the integration of PeVFA which approximates the values of multiple EA agents based on their linear policy representation.
> We do not deem $H$-step bootstrapping itself as the novel point of the proposed surrogate fitness.
>
> The secondary point lies in the correction of bias in state distribution of (Wang et al., 2022).
> In (Wang et al., 2022), value estimates of sampled states from the replay buffer (thus containing intermediate states) are averaged as the surrogate fitness, inducing a mismatch in the definition of fitness.
> We use $H$-step bootstrapping from initial states and thus do not make such a mismatch.
>
>
>
> To avoid possible over-claims as mentioned by the reviewer, we have replaced the "novel" with "new" in the revised version and added some explanations to eliminate possible misunderstanding.
>
>
> &nbsp;
>
> >4. [Re: “The performance gains resulting from the proposed modifications (behavior-level genetic operators and fitness evaluation metric) are not very large.”]
>
>
> Major performance gains achieved by ERL-Re$^2$ come from the two-scale representation framework, while the behavior-level GA operators and the surrogate fitness (which are also based on the two-scale representation) further add to it.
>
> According to the results in Figure 7(a), our behavior-level GA operators outperform ERL's operators marginally in Swimmer and significantly in Ant, and outperform PDERL's significantly in both environments.
> In Figure 7(c), our surrogate fitness brings significant speedup (in the sense of sample efficiency) in Swimmer and marginal speedup in Humanoid.
>
> &nbsp;
>
> >5. [Re: "It is not accurate to call it $H$-step “TD” as there is no temporal difference involved."]
>
>
> We appreciate the reviewer for pointing out this, we agree that the use of "TD" is inaccurate in expression.
> We have corrected $H$-step TD to $H$-step bootstrapping (estimate) in the revised version.
>
>
> ------
>
> We hope our replies have addressed all the concerns the reviewer posed and shown the improved quality of the paper. **We are always willing to answer any of the reviewer's concerns** about our work and we sincerely wish the reviewer to value the technical innovation and overall contributions of the paper.
> We are looking forward to more inspiring discussions.

---

> > ### Author Response · Authors · 2022-11-30
> > **Further Discussions**
> >
> > Dear Reviewer vR7r,
> >
> > Thank you again for reviewing our paper. As the discussion phase has been going on for some time, we'd like to ask one more time if there are further clarifications/modifications about our paper that you like us to address, or if any of the responses to your questions need further elaboration.
> >
> >
> > We also try our best to make the experiments more comprehensive. Specifically, we have added experiments on SMAC (in Appendix D.6 and Figure 25) and on DMC (in Appendix D.5 and Table 3). The results show that ERL-Re$^2$ significantly improves the basic algorithms (i.e., FACMAC and RAD) and outperforms other baselines, which further demonstrates the effectiveness of ERL-Re$^2$.
> >
> >
> > If you believe that our response and revision have addressed your concerns properly, please re-evaluate the contributions of our paper.
> >
> > Thank you,
> >
> > Paper 931 Authors

---

> ### Author Response · Authors · 2022-11-14
> **Initial Response to Reviewer vR7r (Part 1/2)**
>
>
> We sincerely appreciate the reviewer's careful review and constructive comments, and we would like you to know that your questions provide considerably helpful guidance to improve the quality of our paper.
>
> We will try our best to address each of the concerns and questions raised by the reviewer below:
>
> >1. [Re: "Since the algorithm combines elements of RL and EA, it has many hyperparameters to tune."]
>
> Most hyperparameters are innate to the EA and RL base algorithms in ERL-Re$^2$. Concretely, ERL-Re$^2$ is implemented based on the official code of PDERL (using the ERL baseline) and TD3 (including DDPG code), and **all the innate hyperparameters of EA and RL are kept consistent, except eval\_num = 1, pop\_size = 5, elite\_rate = 0.2 are set for all the environments in ERL-Re$^2$**
> (these three hyperparameters are task-dependently tuned in ERL and PDERL, ERL-Re$^2$ avoids tuning them).
>
> For the hyperparameters introduced by ERL-Re$^2$, we show that the algorithm performance is not very sensitive to these hyperparameters and different hyperparameters in proper ranges have similar results that are shown in section 6.3 with Figures 6, 7 and Appendix D.1 with Figures 11, 12 13.
> If the researcher does not want to adjust the hyperparameters, comparative performance can be obtained by setting $\beta$ to 1.0, $p$ to 1.0 and $K$ to 1 across tasks.
> We provide the details in Appendix E.3 for possible help.
>
> &nbsp;
>
> >2. [Re: "I do not feel that I am familiar enough with the EA literature to evaluate the novelty of the proposed behavior-level crossover and mutation."]
>
> To the best of our knowledge of ERL literature, most previous GA operators, i.e., crossover and mutation (or similarly noise perturbation often used in CEM) are conducted at the level of policy (network) parameters, e.g., typical $k$-point crossover and parameter-wise Gaussian noise (as introduced briefly in Section 2).
> As discussed in the paper, this leads to uncontrollable changes to the output of multiple action dimensions.
>
> For some possible alternatives to behavior-level GA operators in ERL literature, GPO (Gangwani et al., 2018) and PDERL (Bodnar et al., 2020) adopt policy (behavior) distillation for crossover operation (i.e., basically behavior clone from parents);
> PDERL proposes a proximal noise for mutation operation.
> For policy-distillation-based crossover, it may be effective in inheriting behaviors during distillation training but interference and approximation are inevitable to DNN, thus having no guarantee on the policy behavior after crossover.
> For proximal mutation noise, it only helps in preventing policy collapse while having nothing to do with behavior-level mutation.
>
> In contrast, based on our two-scale representation framework, our proposed behavior-level GA operators ensure exact effects on policy behavior.

---

### Official Review · Reviewer_eTSz · 2022-11-03

**Confidence:** 4
**Clarity, Quality, Novelty And Reproducibility:** The idea has some novel components as…
**Correctness:** 3
**Technical Novelty And Significance:** 3
**Empirical Novelty And Significance:** 1
**Recommendation:** 5

**Strength And Weaknesses:**

### Strengths

* The general direction of combining evolution and RL seems promising, and maybe under explored despite many similar works in the past few years.
* This paper proposes a slightly different approach, keeping track of two separate policies, and some tricks to share data that may be useful for others.
* The empirical gains are strong in MuJoCo and ablations seem convincing that the method is not overly sensitive.
* It is good to see a limitations section even though it is 80% just discussing future work.

### Weaknesses

* This would have been a good paper in 2019, but right now the field has moved on so far from just showing gains on MuJoCo to provide a useful contribution. Since 2020 it has become common practice to use pixel based tasks, which is no longer expensive to run. Further, there has been a shift in focus towards generalization of agents (see Kirk et al 2021 for a survey), and tasks requiring more complex behaviors. Why does this matter? It is very hard to know if the insights in this work will translate to settings at the forefront of research, and thus the impact seems limited.
* For a pure methods paper, which is solely based on experimental results, I would expect to see at least two distinct domains. Having both TD3 and DDPG doesn't seem additive since they are essentially the same algorithm with a few tricks. This seems like a bit of a duplication which superficially makes the method look more general.
* The method seems to rely heavily on a linear behavior representation. How does this work if we move to a higher dimensional setting? We already know from Mania 2018 that a linear policy can solve MuJoCo, but how about something higher dimensional?
* There is no intuitive analysis as to why this method outperforms where it does. Why is it so good on Swimmer? More broadly, what can we take away from this other than knowing it is great on MuJoCo from proprioceptive states with a 1M timestep budget?

Minor Issues:
* There are a few places where acronyms should be in the parentheses with the reference, e.g. (RL, Sutton 1998). The paper has it like (RL) (Sutton, 1998).
* There are some grammatical mistakes for example "Evolutionary Algorithm" should be plural.
* In the DDPG figure the Swimmer plot has the wrong title.

**Summary Of The Paper:**

Following on from the direction proposed by ERL at NeurIPS 2018, this work seeks to combine evolution and off policy RL into a hybrid algorithm, achieving the best of both. This direction remains promising albeit a little saturated after several years, with many variants proposed. The novel component of this work is to maintain two separate policies which share a representation space, and evolve the evolution policy in the behavior space. Empirical results show gains in the MuJoCo suite for both TD3 and DDPG.

**Summary Of The Review:**

This paper presents a method for general RL tasks. Since the experiments are toy MuJoCo domains, with no analysis over how or why it has gains there, it is hard to know if it is useful for problems we actually care about in the RL community. I thus vote to reject the paper based on the breadth and depth of the experimental results.

---

> ### Author Response · Authors · 2022-11-14
> **Initial Response To Reviewer eTSz (Part 5/5)**
>
>
> >9. [Re: Minor issues]
>
> Thanks for pointing out the minor issues. We have amended them accordingly in the revised version.
>
>
>
> &nbsp;
>
> **Reference:**
>
> [1] Horia Mania, Aurelia Guy, Benjamin Recht. Simple random search of static linear policies is competitive for reinforcement learning. NeurIPS 2018
>
> [2] Francesco Faccio, Louis Kirsch, Jürgen Schmidhuber. Parameter-Based Value Functions. ICLR 2021
>
> [3] Mikayel Samvelyan, Tabish Rashid, Christian Schröder de Witt, Gregory Farquhar, Nantas Nardelli, Tim G. J. Rudner, Chia-Man Hung, Philip H. S. Torr, Jakob N. Foerster, Shimon Whiteson. The StarCraft Multi-Agent Challenge. AAMAS 2019
>
> [4] Bei Peng, Tabish Rashid, Christian Schröder de Witt, Pierre-Alexandre Kamienny, Philip H. S. Torr, Wendelin Boehmer, Shimon Whiteson. Facmac: Factored multi-agent centralised policy gradients. NeurIPS 2021
>
> [5] Tabish Rashid, Mikayel Samvelyan, Christian Schröder de Witt, Gregory Farquhar, Jakob N. Foerster, Shimon Whiteson. QMIX: Monotonic Value Function Factorisation for Deep Multi-Agent Reinforcement Learning. ICML 2018
>
> [6] John Schulman, Filip Wolski, Prafulla Dhariwal, Alec Radford, Oleg Klimov. Proximal Policy Optimization Algorithms. arXiv:1707.06347, 2017
>
> [7] James Queeney, Yannis Paschalidis, Christos G. Cassandras.
> Generalized Proximal Policy Optimization with Sample Reuse. NeurIPS 2021
>
> [8] Matteo Hessel, Joseph Modayil, Hado van Hasselt, Tom Schaul, Georg Ostrovski, Will Dabney, Dan Horgan, Bilal Piot, Mohammad Gheshlaghi Azar, David Silver. Rainbow: Combining Improvements in Deep Reinforcement Learning. AAAI 2018
>
> [9] Raj Ghugare, Homanga Bharadhwaj, Benjamin Eysenbach, Sergey Levine, Ruslan Salakhutdinov. Simplifying Model-based RL: Learning Representations, Latent-space Models, and Policies with One Objective. arXiv:2209.08466, 2022
>
> [10] Ge Yang, Anurag Ajay, Pulkit Agrawal. Overcoming The Spectral Bias of Neural Value Approximation. ICLR 2022
>
> [11] Kei Ota, Tomoaki Oiki, Devesh K. Jha, Toshisada Mariyama, Daniel Nikovski. Can Increasing Input Dimensionality Improve Deep Reinforcement Learning? ICML 2020
>
> [12] Xinyue Chen, Che Wang, Zijian Zhou, Keith W. Ross. Randomized Ensembled Double Q-Learning: Learning Fast Without a Model. ICLR 2021
>
> [13] Will Dabney, André Barreto, Mark Rowland, Robert Dadashi, John Quan, Marc G. Bellemare, David Silver.
> The Value-Improvement Path: Towards Better Representations for Reinforcement Learning. AAAI 2021
>
> [14] Aviral Kumar, Rishabh Agarwal, Tengyu Ma, Aaron C. Courville, George Tucker, Sergey Levine.
> DR3: Value-Based Deep Reinforcement Learning Requires Explicit Regularization. ICLR 2022
>
> [15] Jean Harb, Tom Schaul, Doina Precup, Pierre-Luc Bacon.
> Policy Evaluation Networks. arXiv:2002.11833, 2020
>
> [16] Min Zhang, Hongyao Tang, Jianye Hao, Yan Zheng:
> Towards A Unified Policy Abstraction Theory and Representation Learning Approach in Markov Decision Processes. arXiv:2209.07696, 2022

---

> > ### Comment · Reviewer_eTSz · 2022-11-15
> > **Thank you for your response!**
> >
> > First a meta-point. This response is incredibly long, which dilutes the points you are trying to make and makes it hard as a reviewer to understand the important new information.
> >
> > Second I think you totally missed the point of what I was saying about ARS. It wasn't at all that you should compare against it, just that your experiments are in a toy domain which can be solved by a linear policy, so I don't think its sufficient and I also don't think the linear representation you use will generalize.
> >
> > Finally, your "large scale" experiments on DMC + a few of the simpler SMAC maps are not convincing since you don't compare against any of the ERL baselines. You are essentially just re-showing the ERL result from several years ago, we already know evolution + gradients is more effective than the two in isolation.
> >
> > I am not at all convinced this paper makes a meaningful contribution and I am confidently sticking with my recommendation for rejection.

---

> > > ### Author Response · Authors · 2022-11-16
> > > **Thanks and Further Clarifications/Discussions (Part 2/2)**
> > >
> > > > **[To "You are essentially just re-showing the ERL result from several years ago, we already know evolution + gradients is more effective than the two in isolation"]**
> > >
> > > Objectively speaking, **none of the prior ERL methods considered in our experiments** (i.e., ERL, PDERL, CERL, CEM-RL)) **is more effective than the two in isolation in all the MuJoCo tasks**, as shown in Figures 4 and 5.
> > > Concretely, the EA baseline is not surpassed by other ERL baselines in *Swimmer*; the RL baselines (i.e., DDPG, TD3 and SAC) perform well in *Walker2d*, *Humanoid* and *Ant*.
> > > One key point of our empirical results is, **ERL-Re$^2$** outperforms the ceiling of EA and RL (informally, $max${$EA, RL$}) while the other prior methods blend the performance of EA and RL (informally, $(RL + EA) / 2$) in MuJoCo tasks.
> > > We deem it as a major experimental contribution of this work.
> > >
> > > &nbsp;
> > >
> > > We sincerely appreciate the reviewer's comments again and we are looking forward to further discussions.
> > >
> > > &nbsp;
> > >
> > > **Reference:**
> > >
> > > [1] Arsenii Kuznetsov, Pavel Shvechikov, Alexander Grishin, Dmitry P. Vetrov. Controlling Overestimation Bias with Truncated Mixture of Continuous Distributional Quantile Critics. ICML 2020
> > >
> > > [2] Xinyue Chen, Che Wang, Zijian Zhou, Keith W. Ross. Randomized Ensembled Double Q-Learning: Learning Fast Without a Model. ICLR 2021
> > >
> > > [3] James Queeney, Yannis Paschalidis, Christos G. Cassandras. Generalized Proximal Policy Optimization with Sample Reuse. NeurIPS 2021
> > >
> > > [4] Cristian Bodnar, Ben Day, Pietro Lió. Proximal Distilled Evolutionary Reinforcement Learning. AAAI 2020
> > >
> > > [5] Yuxing Wang, Tiantian Zhang, Yongzhe Chang, Bin Liang, Xueqian Wang, Bo Yuan.
> > > A Surrogate-Assisted Controller for Expensive Evolutionary Reinforcement Learning. Information Sciences, 2022
> > >
> > > [6] Somdeb Majumdar, Shauharda Khadka, Santiago Miret, Stephen McAleer, Kagan Tumer. Evolutionary Reinforcement Learning for Sample-Efficient Multiagent Coordination. ICML 2020

---

> > > ### Author Response · Authors · 2022-11-16
> > > **Thanks and Further Clarifications/Discussions (Part 1/2)**
> > >
> > >
> > > We appreciate the reviewer very much for timely feedback and we sincerely respect the reviewer's opinions.
> > >
> > > For the length of our response, we want to present our main opinions (which are highlighted) with detailed supports and explanations for a thorough discussion, as we cherish this opportunity very much.
> > >
> > >
> > > >**[About ARS and Sample Efficiency]**
> > >
> > > Our main point is not to show the performance advantage of ERL-Re$2$ over ARS, but to clarify our opinion on **sample efficiency**.
> > > **If we IGNORE the sample efficiency** (i.e., ARS which needs over 1e6, 1e7 timesteps is about 1- or 2-order less sample efficient than ERL-Re$^2$ except for Swimmer), **we would agree** that MuJoCo tasks (except for HalfCheetah) can be solved by ARS (or a linear policy).
> > > However, since sample efficiency is one of the most fundamental drawbacks of DRL in our opinion,
> > > to be honest, **we may not be able to agree with "MuJoCo tasks can be solved by ARS (or a linear policy)"**.
> > > Moreover, if we follow the reviewer's opinion on MuJoCo tasks, a confusing question comes to us: since a linear policy can solve MuJoCo tasks in 2018, why do the following works (e.g., TQC [1], REDQ [2], GePPO [3], PDERL [4], SERL [5]) still try to improve the sample efficiency with non-linear policies in toy MuJoCo tasks?
> > >
> > > > **[About Linear Policy Representation]**
> > >
> > > We just realized whether the reviewer misunderstood our linear policy representation.
> > > **Our policies (in a conventional view) are** based on two-scale representation-based policy construction, thus are **non-linear** due to the composition of non-linear shared state representation and linear policy representation.
> > > In our opinion, we **DO NOT** consider a linear policy itself sufficient to solve MuJoCo tasks and generalizable to more complex tasks.
> > > For some possibly verbose clarification, we mentioned that the essence of two-scale representation-based policy construction is the assignment of approximation (or expressivity) ability among the two-scale representations in the reply to Question 7 in the initial response.
> > >
> > > > **[About Generalizability of ERL-Re$2$]**
> > >
> > > After trying to understand the reviewer's comments "I don't think its sufficient and I also don't think the linear representation you use will generalize" and "your 'large scale' experiments on DMC + a few of the simpler SMAC maps are not convincing since you don't compare against any of the ERL baselines",
> > > **we assume that the reviewer recognizes our empirical supports of the generalizability of ERL-Re$^2$ in DMC and SMAC, while the remaining concern is that we only provide ERL-Re$^2$ and EA/RL baselines in DMC and SMAC.**
> > > We have **added the results of a multi-agent ERL method (MERL [6]) as a baseline in SMAC** and we have also **added two more SMAC environments** (i.e., *2s3z* and *3s_vs_3z*}) in Figure 25 of the new revision we uploaded.
> > > We observe MERL performs better than FACMAC in considered SMAC environments expect a significant drop in MMM2; while **ERL-Re$^2$ still outperforms MERL significantly in all the environments (except marginally in *2c_vs_64zg*)**.
> > > We are also running the additional experiments for other ERL baselines, although NONE of the ERL-related works we include in this paper (i.e., mainly ERL, PDERL, CERL, CEM-RL) have been applied in DMC and SMAC.

---

> ### Author Response · Authors · 2022-11-14
> **Initial Response To Reviewer eTSz (Part 4/5)**
>
> > 7. [Re: More discussions on the dependence on linear policy representation]
>
> In essence, the key point of the two-scale representation framework is an assignment of approximation (or expressivity) ability.
> As we mention in our paper, "Thanks to the expressivity of the shared state representation,
> the individual policy representation can have a simple linear form", this means that **linear policy representation is not a mandatory requirement to ERL-Re$^2$, but rather a convenient and effective choice**.
>
> Toward more complex problems where only highly expressive policy can work,
> we may have a few considerations to deal with the circumstances:
>
> (1) To enhance the expressivity of the shared state representation, we think advanced ideas in self-supervised (or pre-training) state representation are promising in achieving this purpose.
> By enhancing the expressivity, we may still be able to use the convenient linear policy representation (actually this is also regarded by recent state representation studies in RL [13,14]).
>
> (2) If the expressivity of the shared state representation could not be enhanced to an adequate degree, we may alter the linear policy representation to non-linear policy layers.
> In this case, two following problems are the realization of behavior-level GA operators and policy representation input used by PeVFA.
> One possible solution to the former problem is to compute the parameter gradients regarding specific action dimension(s) and perform GA operators in the orthogonal directions. This is actually an approximate generalization of the exact behavior-level GA operators with linear policy representation.
> For the second problem, we can resort to layer-wise parameter encoding for policy networks proposed in the PeVFA paper. There are also some alternative choices like [15,16]
>
>
> Regarding optimizing the shared state representation and estimating the fitness,
> the linear policy representations are encoded as an input to PeVFA, which has less overhead compared to encoding the entire network. In Appendix E.2, we describe this encoding process in detail, mainly by encoding each action dimension separately and subsequently summing and averaging them, which is dimension-insensitive and thus can be used to solve high-dimensional problems.
> Our experiments also show that for the higher-dimensions Humanoid task as well as the tasks in SMAC prove that the approach is effective. Of course, we agree that it may be further improved by designing more efficient encoding methods for high-dimensional problems, but this is not the focus of the current work.
>
>
>
>
>
> &nbsp;
>
> > 8. [Re: Others]
>
> (1) For "The novel component of this work is to maintain two separate policies which share a representation space ..." and "keeping track of two separate policies", we maintain five (i.e., size of the EA population) plus one (i.e., the RL agent) policies rather than two policies.
> We assume that the reviewer is aware of this and the "two policies" in the comments means the same thing.
>
> (2) For "... despite many similar works in the past few years", we would appreciate a lot if the reviewer could provide some concrete related works that we may neglect in our paper and point out the similarities and connections. This will strengthen our paper a lot.

---

> ### Author Response · Authors · 2022-11-14
> **Initial Response To Reviewer eTSz (Part 3/5)**
>
> >6. [Re: Intuitive/concrete analysis and explanations on Swimmer and More \& "There is no intuitive analysis as to why this method outperforms where it does"]
>
> We provide some intuitive analysis and explanations of why ERL-Re$^2$ is more effective with associated experiments in Appendix D (mainly presented by **Exp 6,7,8**. Note that they are provided in our initial submission version.
>
> To be specific:
>
>
> (1)  In Exp 7 and Figure 17,
> we compare the empirical percentages of the RL agent being selected as the elite and discarder of the population for ERL-Re$^2$(TD3), ERL(TD3), and PDERL(TD3), in Swimmer.
> The results show that the RL agent in ERL-Re$^2$(TD3) has a significantly higher percentage to be selected as an elite and a lower percentage to be selected as the discader.
> This means **the RL agent in ERL-Re$^2$(TD3) participates or contributes more during the learning process.**
> As widely known, Swimmer is a task where EAs solve well and RL algorithms perform poorly.
> This explains why the RL agents in ERL(TD3) and PDERL(TD3) make fewer contributions to lifting the policy performance.
> In contrast, the RL agent in ERL-Re$^2$(TD3) benefits from the useful and profuse knowledge conveyed by the shared state representation, which allows the RL agent to optimize its linear policy representation in a favorable policy space (and also optimize the shared state representation meanwhile).
> Therefore, the sharing of state representation prevents the RL agent from being stuck at suboptimal policies discovered by solely RL optimization.
>
> (2) In Exp 6 and Figure 15, we show that the EA algorithm (i.e., the typical GA) itself (without RL) can be significantly improved by sharing state representation in HalfCheetah and Ant, where EA can not achieve a high score.
> This indicates that state representation learning is significant to boost the optimization efficiency of EA agents;
> in another word, fitness or evaluation signals can be fully utilized to shape the state representation (thus the policy search/optimization space).
> We consider this reveals promising results by decomposing the learning process of EAs and may be of independent interest.
>
>
> (3) In Exp 8 and Figure 17,
> we show the changes of the maximum, minimum, and average performance of the population after shared state representation learning at each iteration.
> The results show **the updates of shared state representation lift the overall performance of the population**, with a major improvement on inferior individuals.
> This empirically elaborates the effectiveness of shared state representation update towards an overall better direction,
> thus facilitating the evolution of the population.
>
>
> As to the behavior-level GA operators, they have an exact guarantee on the behavior semantics of the perturbed agents,
> since the change made to specific action dimension(s) is controllable.
> For previous GA operators (or random perturbations) that perform at the parameter level, their fragility has been demonstrated in GPO (Gangwani et al., 2018) and PDERL (Bodnar et al., 2020) as we mention in the paper.
>
>
> We hope the experiments we have in the appendix would be helpful in providing useful explanations and intuitive analysis to the reviewer.
> We are looking forward to more inspiring discussions.

---

> ### Author Response · Authors · 2022-11-14
> **Initial Response To Reviewer eTSz (Part 2/5)**
>
> >3. [Re: Additional Results on tasks requiring more complex behaviors, higher-dimensional settings]
>
>
> To address the reviewer's concern, we choose **StarCraft II micromanagement (SMAC) benchmark** [3] as an additional testbed, for its rich environments and high complexity of control.
> The SMAC benchmark requires learning multiple cooperative policies in a large action space to win the combat.
> Details are provided in Appendix D.6 in the revised version.
>
> Concretely, we realize ERL-Re$^2$ upon the advanced Multi-Agent RL algorithm FACMAC [4] and evaluate it on four tasks where three of these tasks have a joint action dimension of over 100.
> The results in Figure 25 show that ERL-Re$^2$(FACMAC) outperforms FACMAC in both efficiency and the final performance by large margins.
> This demonstrates that ERL-Re$^2$ can be applied to the problem with complex behavior space and bring significant improvements.
> We consider this is a promising direction
> to broaden the boundaries of ERL and deliver the effectiveness of ERL-Re$^2$ in MARL, including but not limited to single-agent settings.
>
>
> Indeed, what types of tasks to solve is orthogonal to our approach, we have demonstrated it experimentally by realizing ERL-Re$^2$ based on different representative algorithms and evaluating on different popular benchmarks.
>
>
> &nbsp;
>
> >4. [Re: "I would expect to see at least two distinct domains. Having both TD3 and DDPG doesn't seem additive since they are essentially the same algorithm with a few tricks. This seems like a bit of a duplication which superficially makes the method look more general."]
>
>
> We adopt DDPG/TD3 algorithms and MuJoCo benchmark for the purpose of being consistent with the previous works in ERL literature.
>
> For evaluating ERL-Re$^2$ among multiple algorithm domains,
> we have demonstrated the positive results of realizing ERL-Re$^2$ based on DDPG/TD3 (typical DPG algorithms), SAC (i.e., RAD) and FACMAC (MA DPG algorithm with a value-mixing centralized critic, i.e., QMIX [5]).
> We think we have covered major algorithm domains in off-policy continuous control.
>
> For possibly more complete coverage of algorithm domain,
> we will consider to realizing ERL-Re$^2$ based on PPO [6] or GePPO [7] (a generalized off-policy variant).
> Another potential future direction is to realize ERL-Re$^2$ based on the value-based method, e.g., Rainbow [8].
>
>
> &nbsp;
>
>
> >5. [Re: On Single-task Sample Efficiency v.s. Cross-task Generalization]
>
>
> We appreciate the reviewer's valuable opinion and we do agree with the reviewer that the generalization of RL agents across multiple tasks or variants (as discussed in the survey the reviewer mentioned) is a very fundamental and significant problem in the RL community.
> In this work, we focus on improving the sample efficiency of learning in single tasks.
>
> Poor sample efficiency is a widely known drawback of RL (and maybe more notorious for EA).
> We think the sample efficiency of current DRL algorithms is not satisfactory in a general view, especially when considering practical problems.
> To this end, model-based RL [9], representation learning [10,11], TD bias control [12] and many other directions are still pursuing higher sample efficiency of learning in single tasks, which is also the main purpose of ERL literature.
> In this sense, our proposed method ERL-Re$^2$, achieves higher (or comparable) sample efficiency and learning performance than its EA or RL base algorithms, which is not achieved by previous ERL methods to our knowledge.
>
> Although some positive results have been obtained in DMC visual control tasks (shown in Table 3), we will further develop and evaluate our idea on more visual-input tasks, where extra considerations are required for visual representation learning.
> For cross-task generalization, we also plan to incorporate advanced ideas in learning or pre-training common and generalizable representation to empower ERL-Re$^2$ to generalize across multiple variants (e.g., environmental dynamics, reward functions and goals).

---

> ### Author Response · Authors · 2022-11-14
> **Initial Response To Reviewer eTSz (Part 1/5)**
>
>
> We appreciate the reviewer's valuable review and constructive comments, and we would like you to know that your questions provide considerably helpful guidance to improve the quality of our paper.
>
> We will try our best to address each of the concerns and questions raised by the reviewer below:
>
> >1. [Re: "We already know from Mania 2018 that a linear policy can solve MuJoCo ..."]
>
> After carefully checking the details in ARS [1], i.e., (Mania et al., 2018),
> we found the results below.
> In a nutshell, ERL-Re$^2$ achieves higher performance than ARS in HalfCheetah, Walker2d and Ant (regarding average evaluation), and **ERL-Re$^2$ are one- or two-order more sample-efficient than ARS on the MuJoCo tasks except for Swimmer (and maybe three-order more sample-efficient in Humanoid)**.
>
> (Note the results reported in the main body of their paper (in Table 1) are evaluated in the number of episodes.
> However, the number of timesteps (i.e., interaction steps) in each episode can be different, except for Swimmer and HalfCheetah (with a consistent episode horizon of 1000 timesteps).
> Besides, we are aware of the difference in MuJoCo version, i.e., v1 used in ARS and v2 used in our work.
> To our knowledge, the performance of typical algorithms like DDPG, TD3 and SAC does not differ much in v1 and v2.)
>
> To be concrete:
>
> (1) **ARS does not solve HalfCheetah and Ant**.
> For HalfCheetah, in their Table 1, ARS (their best variant) achieves 3430 after 1707 episodes (i.e., 1.707 * 1e6 timesteps);
> in their Table 4 (in appendix), ARS achieves 2345 after 1e6 timesteps; in their Table 5, ARS achieves 5024 after 1e7 timesteps.
> For Ant, ARS achieves 2072 after 1e7 timesteps in their Table 5 and achieves about 4000 after 3 * 1e7 timesteps (estimated according to their results) in their Figure 3 (in appendix).
> In contrast shown by our Figure 4, in HalfCheetah, ERL-Re$^2$ achieves over 4000 after 1e5 timesteps and over 10000 after 1e6 timesteps; in Ant, ERL-Re$^2$ achieves over 4000 after 3 * 1e5 timesteps and over 6000 after 1e6 timesteps.
>
>
> (2) **ARS solves Hopper and Walker2d, but not in a sample-efficient manner**.
> In their Table 4 (in appendix), ARS achieves 3047 in Hopper after 1e6 timesteps and 896 in Walker2d after 1e6 timesteps.
> In their Table 3 (in appendix), ARS achieves 3403 in Hopper after 2 * 1e6 timesteps and 3830 in Walker2d after 8.14 * 1e6 timesteps.
> In contrast, ERL-Re$^2$ achieves over 3400 in Hopper after 4 * 1e5 timesteps and over 5000 after 5 * 1e5 timesteps (in our Figure 4).
>
>
> (3) **ARS solves Swimmer efficiently and is comparably efficient to ERL-Re$^2$**.
> In their Table 1, ARS (v1) achieves 325 after 100 episodes (i.e.,  1e5 timesteps) and ARS (v2) achieves 325 after 427 episodes (i.e.,  4.27 * 1e5 timesteps).
> In contrast, ERL-Re$^2$ achieves about 325 after 2 * 1e5 timesteps (in our Figure 4).
>
> (4) For Humanoid, we did not find sufficient details to exactly calculate the sample efficiency of ARS in Humanoid.
> According to their Table 1, ARS achieves 6000 after 142600 episodes (i.e., optimistically about 1e8 timesteps).
> In contrast, ERL-Re$^2$ achieves about 6000 after 7 * 1e5 timesteps (in our Figure 4).
>
> The sample efficiency of ARS can also be found in the recent work [2] for a helpful reference.
>
>
>
> In addition, ARS also depends on specific reward shaping (as detailed in their Appendix A.1), although we do not think this is a drawback.
>
>
> &nbsp;
>
> >2. [Re: Results on pixel-based tasks]
>
>
> For evaluating ERL-Re$^2$ on **pixel-based tasks**, we provide the experimental results on DeepMind Control Suite (DMC) in Appendix D.5 and Table 3.
> DMC is a popular benchmark for forefront visual control research.
> **This is provided in our initial submission version** and we note it in section 6.2.
> We assume that the reviewer may have missed it.
>
> Specifically, we realize ERL-Re$^2$ upon RAD (Laskin et al., 2020), i.e., a SAC-based visual control algorithm mainly based on image augmentation, and evaluate ERL-Re$^2$(RAD) on four visual control tasks in DMC.
> We choose RAD due to its simplicity, effectiveness and accessibility to the official code.
> The results demonstrate that **ERL-Re$^2$(RAD) also achieves clear performance improvements over RAD in all the four DMC tasks**.
> This indicates the ability of ERL-Re$^2$ to generalize to other algorithms and the effectiveness in visual control tasks.

---

### Author Response · Authors · 2022-11-14
**Common Response**

We appreciate all the reviewers' valuable and inspiring comments. Individual responses and our revision has been uploaded.

For our **FIRST** revision, major updates are summarized below:


$\bullet$ Additional experimental results are provided, including:

&nbsp;&nbsp;&nbsp;&nbsp;$\circ$ The experiments on high-dimensional complex control tasks StarCraft II micromanagement (SMAC) in Appendix D.6 and Figure 25 (as suggested by **Reviewer eTSz**).
We build ERL-Re$^2$ on FACMAC (Peng et al., 2021). To be concrete, ERL-Re$^2$(FACMAC) realizes the policy of each agent in FACMAC based on our two-scale representation-based policy construction and optimizes the EA and RL agents similarly as done for ERL-Re$^2$ (TD3/DDPG).
More details are provided in Appendix D.6.
We evaluate ERL-Re$^2$(FACMAC) on four tasks, including *2c_vs_1sc (easy)*, *2c_vs_64zg (hard)*, *MMM2 (super hard)* and *so_many_baneling*.
Among these tasks, the dimensions of the joint action on tasks *2c_vs_64zg*, *MMM2* and *so_many_baneling* are all over 100.
The results show that ERL-Re$^2$ (FACMAC) outperforms FACMAC by large margins,
achieving significant improvements with faster convergence and higher final performance on all the tasks.
This indicates the effectiveness and generality of ERL-Re$^2$ in high-dimensional complex control tasks.



&nbsp;&nbsp;&nbsp;&nbsp;$\circ$ The experiments on MuJoCo with 3 million timesteps in Appendix D.7 and Figure 26 (longer training horizon as suggested by **Reviewer ouqq**).
We run 3 million training timesteps for *HalfCheetah*, *Ant*, *Humanoid* and *Walker*, since convergence performance has been achieved for *Hopper* and *Swimmer*.
The results indicate that our algorithm has higher sample efficiency and also consistently better performance during longer training timesteps.

$\bullet$ Some minor issues are fixed: e.g. plural forms, the mismatched title, fixes for set symbols.


-----------------

For our **SECOND** revision, major updates are summarized below:

$\bullet$ Additional experimental results are provided in Appendix D.6 and Figure 25, including two more environments (*3s5z* and *3s_vs_3z*) in SMAC.

$\bullet$ An Multi-agent ERL baseline (MERL [1]) is added to the results on SMAC in Figure 25 (as suggested by **Reviewer eTSz**).

We are running more experiments in DMC and SMAC and trying to provide more complete results.

-----------------

For our **Third** revision, major updates are summarized below:

$\bullet$ Additional experimental results are provided in Appendix D.5 and Table 3, including the ERL-related baselines (i.e., ERL, PDERL, CEM-RL all in RAD version) on DMC. The results show that our algorithm can further improve RAD on pixel-level tasks while other ERL methods fail, which demonstrates the effectiveness and generalization of ERL-Re$^2$ and also demonstrates that each individual of the population independently extracting the useful information from the visual input is very inefficient.


------

**Reference:**

[1] Somdeb Majumdar, Shauharda Khadka, Santiago Miret, Stephen McAleer, Kagan Tumer. Evolutionary Reinforcement Learning for Sample-Efficient Multiagent Coordination. ICML 2020


------



We hope our replies have addressed all the questions and concerns the reviewers posed and shown the improved quality of the paper.
**We are always willing to answer any of the reviewers' concerns** about our work and we sincerely wish the reviewers to value the technical innovation and overall contributions of the paper.
We are looking forward to following inspiring discussions.

---

### Author Response · Authors · 2022-11-17
**Discussion Phase Ending Soon**

Dear Reviewers,

Thank you again for reviewing our paper. As the discussion phase is about to end in one day, we'd like to ask one more time if there are further clarifications/modifications about our paper that you like us to address, or if any of the responses to your questions need further elaboration.

 If you believe that our response and revision have addressed your concerns properly, we sincerely hope that you could re-evaluate the contributions of our paper.


Thank you,

Paper 931 Authors

---

> ### Comment · Reviewer_eTSz · 2022-11-19
> **Stand by my score**
>
> I stand by my score. This paper presents a complex method designed to outperform in MuJoCo. There is no new insight that can transfer, no theoretical underpinning, so the only way this is useful is if the experimental results are very strong. They are not at all - it is just gains on MuJoCo which has not been actively used for RL research for a few years. I can't see how this paper helps the community progress and that is ultimately the most important thing.

---

> > ### Author Response · Authors · 2022-11-19
> > **Thank you for further feedback!**
> >
> > We appreciate the reviewer's further feedback timely and
> > we totally respect your opinions.
> >
> > After another careful attempt in understanding your comments and responses,
> > we have some critical questions below.
> >
> > &nbsp;
> >
> >  First, we are confused about the reviewer's comments "This paper proposes a slightly different approach" (in Strengths of the initial review) and "This paper presents a complex method" (seems to be a weakness).
> >
> > It seems to be kind of **contradictory** to us and we may not be able to get the point of the reviewer precisely.
> > We are wondering **if there is a way or an opportunity that we could provide more useful clarifications on the "slight difference" or the "complexity" of our approach?**
> >
> > It would be a great help if you could provide some concrete explanations on this point.
> >
> > &nbsp;
> >
> > Second, as the reviewer mentioned "the only way this is useful is if the experimental results are very strong",
> > we are also kind of confused about it.
> >
> > For MuJoCo, the reviewer commented "The empirical gains are strong in MuJoCo and ablations seem convincing that the method is not overly sensitive";
> > for other environments and algorithm domains, we also provided the experiments on visual-control environments in DMC in our initial submission (and supplemented several baselines) and added experiments on the SMAC benchmark.
> >
> > According to the newest feedback "They are not at all - it is just gains on MuJoCo which has not been actively used for RL research for a few years",
> > we are wondering **if our results (i.e., the improvements in learning efficiency and performance) on DMC and SMAC mean nothing and provide no useful support to the effectiveness of our method in the reviewer's opinion.**
> > We will appreciate it very much if the reviewer could provide more concrete comments on our results on DMC and SMAC or some suggestions for other "useful" environments and domains.
> >
> > **Or, the reviewer considers that the performance improvements
> > in MuJoCo, DMC and SMAC (that have been achieved recently or are to be achieved in the future) are trivial or negligible to the community progress?**
> >
> >
> > &nbsp;
> >
> > We sincerely appreciate the reviewer's response again and we are looking forward to the reviewer's valuable comments.

---

> > > ### Comment · Reviewer_eTSz · 2022-11-19
> > > **Answers**
> > >
> > > There have been many algorithms combining evolution and off policy RL. You propose another variant, which is fairly complex but the idea of combining the two has been explored before. Hence similar in principle to others, yet complex. Secondly, you do not compare against any of these other methods for DMC and SMAC so those results do not tell us anything about your specific, complex variation on the theme of evolution + off policy RL.

---

> > > > ### Author Response · Authors · 2022-11-19
> > > > **We have added ERL baselines in DMC and SMAC in our second and third revised versions.**
> > > >
> > > >
> > > >
> > > > We appreciate the reviewer's kind explanations.
> > > >
> > > >
> > > >
> > > >
> > > >
> > > >
> > > > > Re: you do not compare against any of these other methods for DMC and SMAC so those results do not tell us anything about your specific, complex variation on the theme of evolution + off policy RL.
> > > >
> > > > **We have added ERL baselines in DMC and SMAC in our second and third revised versions.**
> > > > As in the Common Response:
> > > >
> > > >
> > > > - "A Multi-agent ERL baseline (MERL [1]) is added to the results on SMAC in Figure 25 (as suggested by Reviewer eTSz)."
> > > >
> > > > - "Additional experimental results are provided in Appendix D.5 and Table 3, including the ERL-related baselines (i.e., ERL, PDERL, CEM-RL all in RAD version) on DMC."
> > > >
> > > > We assume that the reviewer may miss the messages. We would like to know if these ERL baselines are useful to the reviewer.
> > > > And we will appreciate it a lot if the reviewer provides more suggestions on concrete useful ERL baselines (existing or feasible in DMC and SMAC).
> > > >
> > > > > Re: You propose another variant, which is fairly complex but the idea of combining the two has been explored before. Hence similar in principle to others, yet complex.
> > > >
> > > > For the two-scale representation-based policy construction, we consider **our shared state representation simplifies the implementation and maintenance of EA population.**
> > > > Also as shown by Table 2 in the appendix, our policy model has fewer parameters when the policy scale is aligned among compared methods.
> > > > In addition, our method is more scalable when the population size scales up due to the sharing.
> > > >
> > > > For the optimization of policies, GA and RL optimize the policies in a linear policy representation space in conventional ways (except for the action-level GA operation), which we think it is not more complex than previous methods (e.g., PDERL needs additional policy distillation for crossover and proximal gradient computation for mutation).
> > > >
> > > > The only additional model introduced is PeVFA, which is used for network approximation of value/fitness of the agents in EA population.
> > > > The structure of PeVFA and the way of training are very similar to a conventional critic/Q function approximator.
> > > >
> > > > However, we respect the reviewer's opinion on the "complexity" and "slight difference" and we will not try to change your opinion on this point.
> > > >
> > > > &nbsp;
> > > >
> > > > Thank you very much again for your response!

---

> > > > > ### Comment · Reviewer_eTSz · 2022-11-19
> > > > > **DMC + Baselines is sufficient for a revision, still some more to do :)**
> > > > >
> > > > > Hi - I did indeed miss the 2nd and 3rd revisions of the paper, only viewing the first.This adds to the meta comment, try to be more concise as it is significantly more effective when reviewers are doing their best but have limited time. For example, you sent me FIVE comments in the reply to the initial review. I don't believe this could not have been done in two.
> > > > >
> > > > > I am increasing to weak reject because I still do not think there is a huge impact from this as it is purely a methods paper without much analysis. That being said, the empirical results now are reasonable. I would encourage the authors to improve along the following dimensions for future revisions:
> > > > >
> > > > > 1) Consider procedurally generated environments: how does this method influence the generalization capability of the agents? This setting is becoming more important in RL (see Kirk 2021 for a survey on generalization). Example environments would be MiniGrid, MiniHack, Procgen, The representations learned in this method might be effective here.
> > > > >
> > > > > 2) More analysis of how and why the changes in this method help. This is not an ablation, but insights that help explain the outperformance. This will benefit you - because if people understand the method they may use and cite it :)
> > > > >
> > > > > 3) Remove one of the two identical MuJoCo experiments in the main body, and switch it with the DMC and SMAC and preferably 1) and 2).
> > > > >
> > > > > Good luck with these changes, they should make the work better!

---

> > > > > > ### Author Response · Authors · 2022-11-22
> > > > > > **We are pleased to address the main concern of the reviewer!**
> > > > > >
> > > > > >
> > > > > > We are pleased that the reviewer recognizes our efforts in the additional experiments and we appreciate the valuable and constructive comments from the reviewer.
> > > > > >
> > > > > >
> > > > > > > Re: Consider procedurally generated environments: how does this method influence the generalization capability of the agents? Remove one of the two identical MuJoCo experiments in the main body, and switch it with the DMC and SMAC and preferably 1) and 2).
> > > > > >
> > > > > >
> > > > > >
> > > > > > We thank the reviewer for the valuable suggestions. We will carefully consider the procedurally generated environment for more complete evaluations in the future.
> > > > > >
> > > > > > For the presentation of experimental results, we will adjust the experimental content in the main text and the appendix according to the reviewer's comments.
> > > > > > In addition, we will also move the analysis on algorithm behaviors (e.g., Exp 6,7,8 in Appendix D) from the appendix into the main text to enhance the intuitive explanations of the effectiveness of our method.
> > > > > >
> > > > > >
> > > > > > &nbsp;
> > > > > >
> > > > > >
> > > > > > > Re: More analysis of how and why the changes in this method help. This is not an ablation, but insights that help explain the outperformance.
> > > > > >
> > > > > > We provide a few different experiments for intuitive/concrete analysis and explanations to the outerperformance of Swimmer and others, in Appendix D (mainly presented by Exp 6,7,8). We highlighted this in our first response (See Point (6)), which the reviewer may not have noticed. We hope that these experiments will be helpful in address the reviewer's remaining concern.
> > > > > > If the reviewer have additional suggestions for experimental analysis, we appreciate further discussion, which would greatly help to improve the paper.
> > > > > >
> > > > > >
> > > > > >
> > > > > > Thank you very much again for your response!

---

### Author Response · Authors · 2022-12-05
**We provide more experiments on MiniGrid**

Dear Reviewer eTSz,

In our last response, we primarily attempt to show why our method is effective by showing more analysis (e.g., Exp 6,7,8 in Appendix D).

Following your suggestion on experiments in procedurally generated environments, we evaluate our method and other baselines on **MiniGrid**. Specifically, we build ERL-Re$^2$, ERL, PDERL, CEM-RL on SAC and evaluate them on three tasks: *DoorKey*, *BoxKey*, *UnlockPickUp*. In *DoorKey*, the agent needs to get the key to open the door to the target location. In *BoxKey*, the agent needs to find the box where the key is located first, open the box to take out the key, and then open the door to reach the target location. In *UnlockPickUp*, the agent needs to find the key to open the door, then drop the key and pick up the box.

These environments are randomly generated by the program, which means the layout of the objects in the grid world is various. We train in *DoorKey* and *BoxKey* with size 8$\*$8, and in *UnlockPickUp* with size 6$\*$6 for 3 million steps; and then we save the trained policies every 200000 steps (during the learning process), and evaluate them on the environments with other sizes (ranging from 6$\*$6 to 20$\*$20, see the table below);
and the best results among the evaluation time points are recorded.

We have tuned the hyperparameters of baselines to provide the best results. For ERL-Re$^2$, we only tune $\beta$ (select from {0.1, 0.05, 0.01}) and set $K=1, p=1.0, H=0$ for all tasks. We evaluate each method 5 times with different seeds and report the mean & standard deviation.

We show the results in the following table:


| | | | | | | |
|-|-|-|-|-|-|-|
| |Size|SAC|ERL|PDERL|CEM-SAC|Ours|
|DooKey|6*6|0.828±0.102|0.880±0.088|0.942±0.050|0.470±0.049|**0.972±0.001**|
| |8*8 (tr)|0.957±0.020|0.937±0.026|0.919±0.098|0.370±0.084|**0.976±0.005**|
| |10*10|0.586±0.129|0.603±0.047|0.758±0.181|0.313±0.119|**0.847±0.053**|
| |12*12|0.328±0.090|0.307±0.047|0.491±0.145|0.227±0.102|**0.573±0.102**|
| |14*14|0.165±0.086|0.201±0.021|0.301±0.109|0.169±0.126|**0.367±0.031**|
| |16*16|0.123±0.067|0.140±0.035|0.174±0.032|0.101±0.049|**0.196±0.046**|
| |18*18|0.073±0.024|0.095±0.018|0.130±0.030|0.087±0.076|**0.146±0.029**|
| |20*20|0.057±0.024|0.074±0.026|0.119±0.041|0.079±0.050|**0.127±0.030**|
|BoxKey|6*6|0.603±0.153|0.640±0.061|0.594±0.150|0.097±0.037|**0.734±0.036**|
| |8*8 (tr)|0.705±0.181|0.748±0.066|0.692±0.143|0.087±0.034|**0.849±0.019**|
| |10*10|0.647±0.210|0.660±0.078|0.589±0.174|0.042±0.011|**0.763±0.029**|
| |12*12|0.432±0.153|0.455±0.075|0.428±0.126|0.036±0.001|**0.596±0.060**|
| |14*14|0.297±0.159|0.301±0.058|0.294±0.105|0.032±0.025|**0.396±0.045**|
| |16*16|0.218±0.087|0.226±0.030|0.210±0.087|0.023±0.006|**0.300±0.086**|
| |18*18|0.125±0.087|0.152±0.018|0.182±0.068|0.020±0.0021|**0.213±0.061**|
| |20*20|0.094±0.064|0.124±0.020|0.125±0.044|0.026±0.015|**0.151±0.037**|
|UnLockPickUp|6*6 (tr)|0.846±0.104|0.812±0.062|0.893±0.017|0.028±0.020|**0.896±0.010**|
| |8*8|0.457±0.131|0.519±0.208|0.441±0.123|0.016±0.010|**0.558±0.038**|
| |10*10|0.107±0.039|0.160±0.108|0.085±0.061|0.008±0.005|**0.172±0.080**|
| |12*12|0.027±0.022|0.049±0.038|0.036±0.022|0.004±0.008|**0.087±0.064**|
| |14*14|0.021±0.015|0.014±0.014|0.010±0.011|0.002±0.006|**0.027±0.033**|
| |16*16|0.004±0.008|0.004±0.008|0.005±0.010|0.000±0.000|**0.010±0.016**|
| |18*18|0.000±0.000|0.003±0.006|0.005±0.010|0.000±0.000|**0.006±0.009**|
| |20*20|0.000±0.000|0.000±0.000|0.000±0.000|0.000±0.000|**0.004±0.006**|



Experimental results show that ERL-Re$^2$ has a significant improvement over SAC and outperforms other algorithms
when evaluated on untrained environments in a zero-shot manner.
This demonstrates that ERL-Re$^2$ can also bring a steady improvement in generalization in such procedurally generated environments.
An intuitive explanation of why ERL-Re$^2$ is more effective is that the learning of shared state representations needs to be beneficial for all policies (maintained by ERL-Re$^2$), and it helps to construct a more general representation space rather than a 'narrow' one specific to a single agent, which is less sensitive to changes in environments and thus more efficient in generalization.
This is also consistent to our empirical analysis of the shared representation conducted in Exp 6,7,8 in Appendix D.

We're also conducting more experiments on MiniHack.

We hope that the reviewer can re-evaluate our work in light of our responses above.

Thank you.

---

> ### Author Response · Authors · 2022-12-13
> **Discussion Phase Ending Soon**
>
> Dear Reviewer eTSz, as the entire discussion phase is about to end. we'd like to ask whether you have any question about the further clarifications and experiments we have provided. If our responses have addressed your concerns properly, please consider increasing your score. Thank you.

---

### Decision · Program_Chairs · 2023-01-20

**Decision:**

Accept: poster

**Justification For Why Not Higher Score:**

The two points above

**Justification For Why Not Lower Score:**

Interesting relatively simple method that has a good performance (though should be tested on more complex environments).

**Metareview: Summary, Strengths And Weaknesses:**

This paper presents a method for evolving a collection of agents which share most of their network and achieves good performance and benchmarks presented. There are two primary things that the paper should improve, the first I hope can be done for the paper and the second for future work.
- It would be good to understand better why the method works and how it operates. For example, as all the agents are optimising the same objective, why don't all W's converge to the same value. Are they close to a single value, with noise equaling the evolutionary perturbation and is it that randomness that improves the exploration, or is it some other mechanism?
- Run it on more complex environments. The environments that this has been run on are short term reactive environments, like walker in continuous control. It is good that the authors extended the set of environments since then, and hope they put it well in the final version of the paper. However, more complex environments with long term learning are really what should be used here.


**Note From Pc:**

if the above contains the word "oral" or "spotlight" please see: "oral" presentation means -> notable-top-5% and "spotlight" means -> notable-top-25%. As stated in our emails, we are disassociating presentation type from AC recommendations